# Efficacy of Wang Nam Yen herbal tea on human milk production: A randomized controlled trial

**Koollachart Saejueng**[1], **Tanawin Nopsopon**[2], **Piyawadee Wuttikonsammakit**[3], **Wattanaporn Khumbun**[4], **Krit Pongpirul**[2,5] *

**1** Department of Obstetrics and Gynecology, Bandung Crown Prince Hospital, Udon Thani, Thailand,
**2** Department of Preventive and Social Medicine, Faculty of Medicine, Chulalongkorn University, Bangkok, Thailand, **3** Department of Obstetrics and Gynecology, Sunpasitthiprasong Hospital, Ubon Ratchathani, Thailand, **4** Department of Thai Traditional and Alternative Medicine, Sunpasitthiprasong Hospital, Ubon Ratchathani, Thailand, **5** Department of International Health, Johns Hopkins Bloomberg School of Public Health, Baltimore, MD, United States of America

* doctorkrit@gmail.com

**Data Availability Statement:** All relevant data are within the paper and its Supporting Information files.

## Abstract

### Background

Insufficient milk production is a common problem affecting breastfeeding women, in particular following Cesarean delivery. Wang Nam Yen herbal tea is a promising traditional Thai medicine used by postpartum women to stimulate milk production, as an alternative to pharmaceutical galactagogues. We aimed to compare the efficacy of Wang Nam Yen herbal tea, domperidone, and placebo, in increasing milk production in mothers who underwent Cesarean delivery.

### Methods

Women who underwent uncomplicated cesarean delivery at Sunpasitthiprasong Hospital were randomized into three groups. The participants received the treatments daily for three consecutive days. The primary outcome was breast milk volume at 72 hours after delivery. Secondary outcomes were pregnancy and neonatal outcomes, adverse events, and participant satisfaction.

### Results

Of the 1,450 pregnant women that underwent cesarean delivery, 120 women were enrolled. Their mean age and gestational ages were 28.7 years and 38.4 weeks, respectively. Breast milk volume at 72 hours postpartum was significantly different among the three groups ($p = 0.030$). The post hoc Bonferroni correction indicated a significant difference in breast milk volume between Wang Nam Yen herbal tea group and placebo control group ($p = 0.007$) while there was no difference between Wang Nam Yen herbal tea group and domperidone group ($p = 0.806$) and between domperidone group and placebo control group ($p = 0.018$). There was no difference in pregnancy and neonatal outcomes, adverse events, and participant satisfaction among the three groups.

**Funding:** The authors received financial support for the research from the Department of Thai Traditional and Alternative Medicine, Ministry of Public Health, Thailand. The funding source had no involvement in research preparation, study design; collection, analysis, interpretation of data; writing of the report; or in the decision to submit the article for publication.

**Competing interests:** The authors have declared that no competing interests exist.

## Conclusion

Wang Nam Yen herbal tea was effective in augmenting breast milk production at 72 hours postpartum in mothers following cesarean delivery, and there was no evidence that herbal tea and domperidone differed in terms of augmenting breast milk production.

## Trial registration

The study was approved by the institutional review board of Sunpasitthiprasong Hospital (No.061/2559) and was registered TCTR20170811003 with the Thai Clinical Trial Registry.

## Introduction

Breastfeeding advantages are well documented and considered the normative and recommended standard for infant nutrition [1, 2]. Women who undergone cesarean deliveries have a risk of inadequate breast milk production [1]. Low production of breast milk is one of the most common causes of inadequate breastfeeding, which could be supported through various techniques, including the use of substances that stimulate the production of breast milk, known as galactagogues [1, 2]. Domperidone is a commonly prescribed galactagogue, which promotes breast milk production in women experiencing low breast milk supply, with low levels of transfer to maternal milk and no substantial alterations in the nutrient composition [3–6]. The lactogenic effect of domperidone could be as early as two hours after ingestion measured by prolactin level [7], while other studies showed significant effects after 24 hours of treatment measured by prolactin level [8], and daily milk volume [9]. Despite no side-effects being reported in infants of mothers who had used domperidone, domperidone was controversial and had never been approved for marketing by the US FDA due to the potential increased risk of ventricular arrhythmia and sudden cardiac death in adults [2, 10, 11].

As an alternative to pharmaceutical products, several complementary and alternative techniques could be used as galactagogues [12]. Herbal medicines and techniques including herbal compression, herbal supplements, and herbal teas containing ginger, stinging nettle, fenugreek, or turmeric, are effective for promoting breast milk production without adverse effects [13–16]. In Thailand, numerous traditional galactagogues, including banana flower, lemon basil, Thai basil, bottle gourd, and pumpkin, have significant correlations with increased breast milk volume [17].

Several factors are associated with inadequate breastfeeding due to low milk production, including cesarean deliveries which can lead to lactation difficulties [1, 2, 9]. Postpartum treatment with Domperidone in women who have undergone full-term cesarean has shown augmented production of breast milk [9]. At Wang Nam Yen hospital in Thailand, the Wang Nam Yen herbal tea, consisting of sappan (*Caesalpinia sappan* Linn.), licorice (*Glycyrrhiza glabra* Linn.), bael fruit (*Aegle marmelos* L. Corr), ginger (*Zingiber officinale* Roscoe), and jewel vine (*Derris scandens* (Roxb.) Benth) have been used as a traditional Thai medicine to stimulate milk production in postpartum women. Wang Nam Yen herbal tea is expected to provide a lactogenic effect as early as 24 hours after first ingested and a significant effect at 72 hours of ingestion as presented in a previously unpublished study measured by daily milk volume [18].

The most active galactagogue ingredient in Wang Nam Yen hospital is ginger which has evidence on lactogenic effect as a single treatment after 48–72 hours of ingestion measured by daily milk volume [16], and as combined treatment with other herbs after 24 hours of

ingestion measured by prolactin level [19]. The other four ingredients are based on Thai traditional medicine knowledge on alleviation of postpartum elemental imbalance caused by losing stamina, water, and blood from delivery and muscle pain [20–22]. Acute oral toxicity assessment of the tea in Wistar rats revealed a median lethal dose (LD50) of 5,000 mg/kg with no cases of serious side effects [18]. While data from the grey literature found a significant increase in breast milk volume with no serious side effects in the herbal tea group compared to placebo, a randomized controlled trial with standard treatment as active control arm was necessary to provide substantive evidence of the herbal tea efficacy in stimulating breast milk production.

The Wang Nam Yen herbal tea is a promising traditional Thai galactagogue, which might be used as an alternative to the common pharmaceutical galactagogue, domperidone, to support breastfeeding difficulties due to low supply, especially in women that have undergone cesarean deliveries [18]. The objective of this study was to compare the efficacy of Wang Nam Yen herbal tea, domperidone, and a placebo, in augmenting breast milk production in mothers that have undergone Cesarean delivery, in a double-blinded randomized controlled trial.

## Methods

### Study design

Tea4Milk was a three-armed, double-blinded randomized controlled trial on human milk production conducted in Ubon Ratchathani Province, Thailand at Sunpasitthiprasong Hospital, a regional hospital under the Ministry of Public Health, from February 2017 to September 2017. Participants were recruited after delivery via cesarean section by placing advertisements in four obstetrics wards. All patients provided written informed consent. The study was reviewed and approved by the institutional review board of Sunpasitthiprasong Hospital (No.061/2559). The trial was registered and kept up to date at the Thai Clinical Trial Registry (Registration No. TCTR20170811003).

### Participants

Inclusion criteria were women aged 15 to 41 years that had delivered via cesarean section at 28 to 42 weeks gestational age. Participants with contraindications for breastfeeding and/or serious illness affecting mother or infant, such as human immunodeficiency virus infection, postpartum hemorrhage with hypovolemic shock, HELLP syndrome (hemolysis, elevated liver enzymes, and low platelet count), eclampsia with respiratory failure, or had a history of allergy to domperidone or ingredients in the herbal tea, or neonates-mother separation before participation were excluded from the study.

### Randomization and blinding

Participants were recruited according to inclusion and exclusion criteria through the advertisements at four obstetrics wards by obstetricians and nurses who did not have information on the allocation of participants. Participants were randomly allocated into 3 groups with a 1: 1: 1 ratio by block randomization with a block of three using the table of random numbers for creating randomization ID which was inserted into an opaque envelop to mask clinicians, data collectors, and patients. Randomization ID was assigned as 1) Wang Nam Yen herbal tea and placebo tablet (T group), 2) domperidone tablet and placebo herbal tea (D group), 3) placebo tablet and placebo herbal tea (C group). When 120 randomization IDs were generated, the ID was joined together starting with a randomization ID for participant number 9 (TD009) as the first ID.

Participants and investigators were blinded to the treatment assignment. After generating randomization codes, the research team responsible for blinding created an opaque envelope to seal the randomization codes. The front end of an opaque sealed envelope presented the study name, principal investigator name, site of study, randomization ID, study ID, and specific date to open envelop. There was a document inside the sealed envelope with randomization ID, assigned group, study ID, specific date and time to open envelop, and signature of envelope opener who was not enrollment personnel nor clinician nor data collector. All study interventions were prepacked and sealed in opaque packages, for three consecutive days of postpartum use.

## Interventions

Wang Nam Yen herbal tea was manufactured by Wang Nam Yen hospital, a community hospital in Sa Kaeo reputed for Thai traditional herbal medicine. Wang Nam Yen herbal tea was packed in a tea bag designed for each meal. Each Wang Nam Yen herbal tea bag contained 500 mg sappan (*Caesalpinia sappan* Linn.), 500 mg licorice (*Glycyrrhiza glabra* Linn.), 500 mg bale fruit (*Aegle marmelos* L. Corr), 500 mg ginger (*Zingiber officinale* Roscoe), and 500 mg jewel vine (*Derris scandens* (Roxb.) Benth). The herbal tea placebo was also prepared as a teabag for each meal, and the placebo tea bag was identical to Wang Nam Yen teabag. The placebo teabag was filled with 2,500 mg pandan (*Pandanus amaryllifolius*) leaves. All herbal ingredients were from Charoensuk Pharma Supply Co., Ltd. Both herbal teas were prepared by diffusion in 200 mL of warm water for 5 minutes which gave an identical appearance. Interventions were administered orally three times per day after meals. The first administration occurred 12–18 hours following delivery to ensure sufficient time for the evaluation of any postpartum complications that may have occurred. Domperidone was given 10 mg 3 times per day (total dose 30mg daily) which was considered safe and sufficient for increasing breast milk production [23, 24]. The placebo tablet was made of glutinous rice flour with an identical appearance to the domperidone tablet. All participants (T, D, C groups) were administered 200 mL of tea and a tablet after each meal. All groups had similar breastfeeding support for early breastfeeding as early as possible, frequent breastfeeding, correct positioning, and avoiding the use of formula when possible. There was no restriction on early breastfeeding before the first dose of intervention. In case mother-infant separation occurred after randomization, the mothers would continue in the trial similar to other participants except direct breastfeeding would not be possible while the infants would receive standard of care.

T–The Wang Nam Yen herbal tea treatment group received Wang Nam Yen herbal tea and placebo tablet after each meal for three meals per day. This was an intervention arm that was designed to assess the efficacy of Wang Nam Yen herbal tea.

D–The domperidone treatment group received an herbal tea placebo and a 10 mg domperidone tablet after each meal for three meals per day. This was an active control arm that was designed to be a comparator as standard treatment.

C–The placebo control group received both herbal tea placebo and a placebo tablet after each meal for three meals per day. This was a placebo control arm that was designed to be a reference group with no treatment.

## Outcome measures

The primary outcome was breast milk volume at 72 hours after delivery. The first breastfeeding was encouraged to be as early as possible and no late than after the first dose of intervention which occurred 12–18 hours following delivery. All participants were encouraged to breastfeed their infants based on Sunpasitthiprasong hospital's standard of care policy. The

breast milk volume was measured with a two-hour interval from the last breastfeeding, using an electronic pump (Spectra S2Plus®, Spectra Baby USA). The electronic pump was applied with help from a trained nurse for 15 minutes during each breast and the total volume of milk was recorded. Breast milk volume was also measured at 24 hours and 48 hours post-delivery. The milk collection was done uniformly for all participants. While some babies might demand hourly feeding, all mothers were notified at 2 hours before each milk collection session which allowed the last breastfeeding before the milk collection. The unit of analysis was the total amount of milk volume in a single 15-minute expression (mL/extraction). The breast milk volume at 24 hours post-delivery was used as the baseline.

Secondary outcomes were observed 72 hours after the delivery period including (1) pregnancy outcomes such as postpartum hemorrhage, endometritis, (2) neonatal outcomes such as neonatal jaundice, respiratory distress, (3) adverse effects including headache, dry mouth, diarrhea, muscle cramps, itching, or allergic reactions assessed every 24 hours by the ward nurses and rechecked by a research assistant, and (4) participant satisfaction during the three days' postpartum period were also recorded.

Fluid intake and output were recorded daily to assess the maternal fluid volume status. The need for formula milk supplementation was recorded daily due to formula milk supplements might affect both maternal breastmilk production and neonatal breastmilk intake.

## Sample size calculation

We used mean and standard deviation of daily breast milk volume (mean $_{treatment}$ = 191.3, mean $_{control}$ = 91.4, standard deviation = 136.1) based on a previous randomized controlled trial by Jantarasaengaram and Sreewapa, 2012 on the effects of domperidone compared to placebo in augmenting lactation following cesarean delivery with 80% power and a 2-sided type I error at 5% [9]. The result required 34.5 samples in each group. Taking into account an expected follow-up loss of 15%, the anticipated sample size of 120 would require a sample size of 40 in each group.

## Statistical analysis

The statistical analysis was performed on an intention-to-treat basis using Stata/MP software version 15.0 (StataCorp 2017, College Station, TX). Descriptive statistics were carried out using mean with standard deviation or median with interquartile range when appropriate. Categorical data were presented as counts and percentages and tested for significance with the Chi-square test. Continuous data were assessed for normality and tested for significance with one-way ANOVA with Bonferroni correction for comparison among groups. Statistical significance was defined as $p < 0.05$ and as $p < 0.017$ with Bonferroni correction.

## Ethics committee approval

The study was approved by the institutional review board of Sunpasitthiprasong Hospital (No.061/2559) and was registered TCTR20170811003 with the Thai Clinical Trial Registry. Written informed consent was obtained from all participants. For participants aged 15–17-year-old, written informed consents were obtained from their parents or guardians.

## Results

### Participant characteristics

The 1450 women who underwent Cesarean delivery in Sunpasitthiprasong hospital from February 2017 to September 2017 were assessed for eligibility. 303 were excluded due to severe

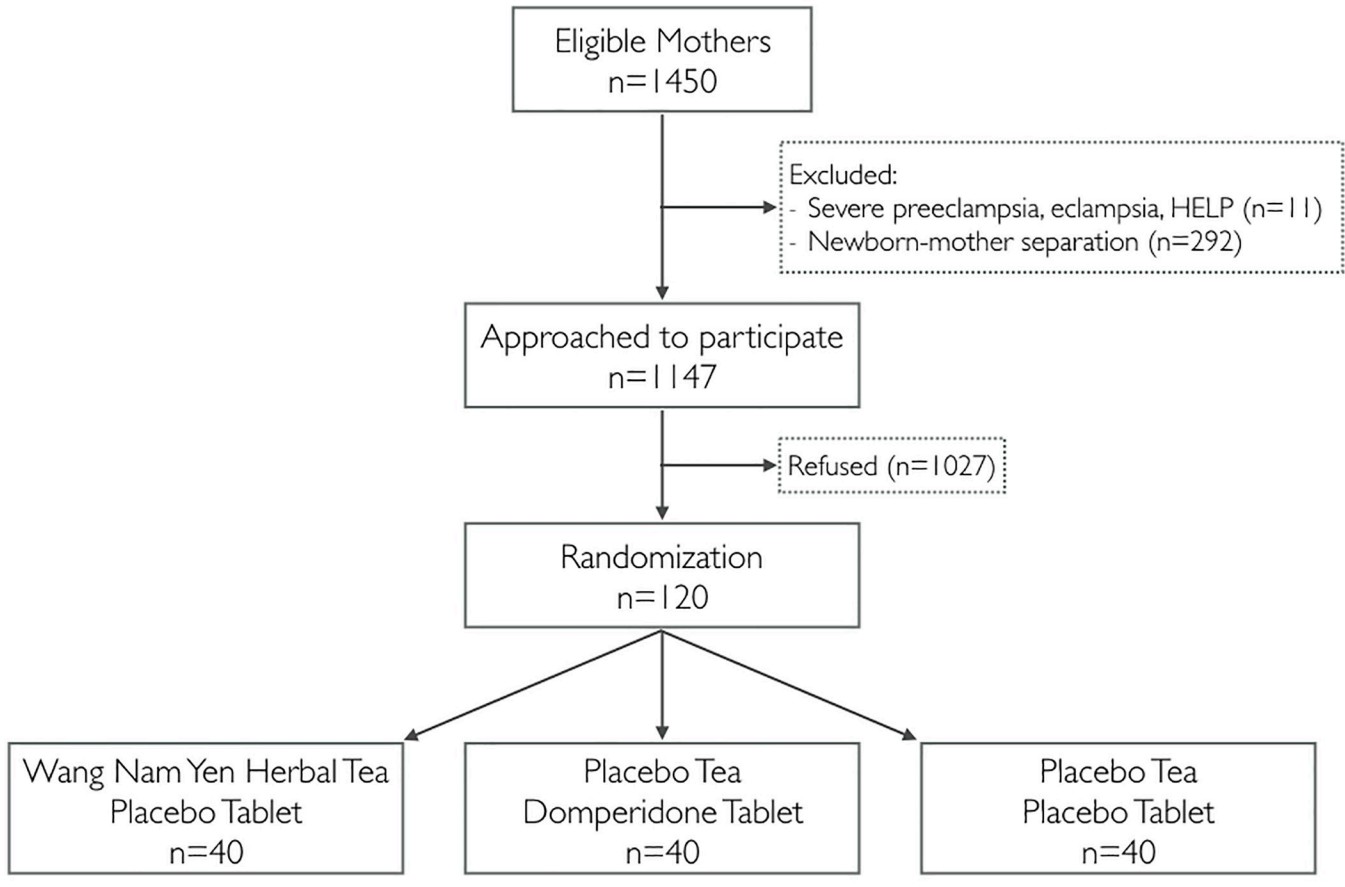

**Fig 1. Flow diagram of the clinical trial.**

eclampsia, eclampsia, HELLP syndrome, or neonatal separation from mother immediately after birth and 1027 declined participation. A total of 120 mother-infant dyads were enrolled in this study and randomized into three groups, with 40 participants in each (Fig 1). There were six participants whose infants required special care in the neonatal intensive care unit but continued to express their milk via breast pump; three were from the control group, two from the domperidone group, and one from the herbal tea group. All participants completed their prescribed intervention and were analyzed based on their randomization group.

The mean participant age was 28.7 years old with an mean gestational age at delivery of 38.4 weeks. There were no substantial differences between the groups in demographic data including maternal age, highest education, method of payment, and salary per month; or in baseline obstetric data including gestational age, parity, the number of antenatal visits, labor symptoms, the indication of cesarean delivery, birth weight, and sex of the baby (Table 1). Additionally, there were similar rates of immediate postpartum hemorrhage defined by blood loss of more than 1000ml, the average fluid intake and output during the 72 hours postpartum period, and the amount of formula milk required (Table 2).

### Primary and secondary outcomes

Breast milk volume per expression at 72 hours postpartum was significantly different among the three groups ($p$ = 0.030). The posthoc Bonferroni correction indicated a significant

**Table 1. Baseline characteristics of the study participants.**

| Baseline data | Total (n = 120) | Herbal tea (n = 40) | Domperidone (n = 40) | Control (n = 40) |
|---|---|---|---|---|
| **Demographic characteristics** | | | | |
| Age (years; mean ± SD) | 28.7 ± 5.9 | 29.2 ± 5.7 | 28.7 ± 5.9 | 28.3 ± 6.2 |
| Monthly income (Baht; median, IQR) (n = 99)* | 5000 (0, 14000) | 5000 (0, 15000) | 3000 (0, 11500) | 3600 (0, 13000) |
| Method of payment, n (%) | | | | |
| Public universal health coverage | 67 (55.8%) | 20 (50.0%) | 27 (67.5%) | 20 (50.0%) |
| Social welfare | 37 (30.8%) | 13 (32.5%) | 11 (27.5%) | 13 (32.5%) |
| Government official's welfare | 8 (6.7%) | 4 (10.0%) | 2 (5.0%) | 2 (5.0%) |
| Other | 8 (6.7%) | 3 (7.5%) | 0 (0.0%) | 5 (12.5%) |
| **Obstetric characteristics** | | | | |
| Gestational age (weeks; mean ± SD) | 38.4 ± 1.5 | 38.6 ± 1.0 | 38.1 ± 2.0 | 38.7 ± 1.1 |
| Primigravida, n (%) | 41 (34.2%) | 12 (30.0%) | 12 (30.0%) | 17 (42.5%) |
| Antenatal visits more than 3 times, n (%) | 58 (48.3%) | 19 (47.5%) | 19 (47.5%) | 20 (50.0%) |
| Presence of labor symptoms, n (%) | 66 (55.0%) | 21 (52.5%) | 23 (57.5%) | 22 (55.0%) |
| Indication of Cesarean delivery, n (%) | | | | |
| Cephalopelvic disproportion | 48 (40.0%) | 15 (37.5%) | 15 (37.5%) | 18 (45.0%) |
| Fetal distress | 8 (6.7%) | 3 (7.5%) | 3 (7.5%) | 2 (5.0%) |
| Abnormal presentation | 9 (7.5%) | 3 (7.5%) | 4 (10.0%) | 2 (5.0%) |
| Previous Cesarean delivery | 48 (40.0%) | 18 (45.0%) | 15 (37.5%) | 15 (37.5%) |
| Others | 7 (5.8%) | 1 (2.5%) | 3 (7.5%) | 3 (7.5%) |
| Experience of breastfeeding, n (%) | 64 (53.3%) | 24 (60.0%) | 21 (52.5%) | 19 (47.5%) |
| **Neonatal characteristics** | | | | |
| Birthweight (grams; median, IQR) | 3067.5 (2737.5, 3337.5) | 3147.5 (2736.3, 3475.0) | 3055.0 (2743.8, 3302.5) | 2975.0 (2715.0, 3307.5) |
| Sex of baby, n (%) | | | | |
| Female | 60 (50.0%) | 20 (50.0%) | 18 (45.0%) | 22 (55.0%) |
| Male | 60 (50.0%) | 20 (50.0%) | 22 (55.0%) | 18 (45.0%) |

IQR, interquartile range; SD, Standard deviation.

*Data not available for all randomized participants.

**Table 2. Puerperal characteristics of the study participants.**

| Puerperal characteristics | Total (n = 120) | Herbal tea (n = 40) | Domperidone (n = 40) | Control (n = 40) |
|---|---|---|---|---|
| Immediate PPH, n (%) | 2 (1.7%) | 1 (0.8%) | 0 (0.0%) | 1 (0.8%) |
| Average fluid intake per day (ml; mean ± SD) | 2566.2 ± 560.6 | 2525.7 ± 436.9 | 2602.9 ± 573.6 | 2569.9 ± 660.4 |
| Average fluid output per day (ml; mean ± SD) | 1826.0 ± 542.7 | 1781.3 ± 452.5 | 1812.6 ± 583.2 | 1884.2 ± 590.0 |
| Formula milk required, n (%) (n = 115)*,**,*** | | | | |
| Not required | 40 (34.8%) | 15 (38.5%) | 12 (31.6%) | 13 (34.2%) |
| Required before 24 hours | 22 (19.1%) | 6 (15.4%) | 9 (23.7%) | 7 (18.4%) |
| Required before 24 hours and 24–48 hours | 22 (19.1%) | 7 (18.0%) | 8 (21.1%) | 7 (18.4%) |
| Required before 24 hours, 24–48 hours and 48–72 hours | 31 (27.0%) | 11 (28.2%) | 9 (23.7%) | 11 (29.0%) |

PPH, postpartum hemorrhage; SD, Standard deviation.

*Data not available for all randomized participants.

**Total percentage is not equal to 100% due to rounding.

***There were no significant differences across the three groups (p = 0.971; Chi-square test).

**Table 3. Primary and secondary outcome measures for maternal complications, neonatal complications, and drug adverse events.**

| Outcomes | Total (n = 120) | Herbal tea (n = 40) | Domperidone (n = 40) | Control (n = 40) | *p* value |
|---|---|---|---|---|---|
| Total milk volume per expression | | | | | |
| Milk volume at 24 hours (ml; mean ± SD) | 5.5 ± 13.9 | 10.0 ± 21.4 | 3.8 ± 7.6 | 2.6 ± 6.2 | 0.036 |
| Herbal vs Placebo | | | | | 0.039 |
| Domperidone vs Placebo | | | | | 0.433 |
| Herbal vs Domperidone | | | | | 0.089 |
| Milk volume at 48 hours (ml; mean ± SD) | 24.5 ± 34.3 | 28.0 ± 39.8 | 28.4 ± 40.8 | 17.2 ± 16.5 | 0.256 |
| Milk volume at 72 hours (ml; mean ± SD) | 50.1 ± 53.9 | 57.5 ± 50.7 | 60.9 ± 70.7 | 31.9 ± 27.7 | 0.030 |
| Herbal vs Placebo | | | | | 0.007 |
| Domperidone vs Placebo | | | | | 0.018 |
| Herbal vs Domperidone | | | | | 0.806 |
| Maternal complications | | | | | |
| Late PPH | 0 (0.0%) | 0 (0.0%) | 0 (0.0%) | 0 (0.0%) | N/A |
| Postpartum endometritis | 1 (0.8%) | 0 (0.0%) | 0 (0.0%) | 1 (2.5%) | 0.365 |
| Neonatal complications | | | | | |
| Neonatal jaundice | 6 (5.0%) | 4 (10.0%) | 0 (0.0%) | 2 (5.0%) | 0.127 |
| NICU admission | 6 (5.0%) | 1 (2.5%) | 2 (5.0%) | 3 (7.5%) | 0.591 |
| Drug adverse events | | | | | |
| Adverse event at 24 hours | 2 (1.7%) | 1 (2.5%) | 1 (2.5%) | 0 (0.0%) | 0.601 |
| Adverse event at 48 hours | 0 (0.0%) | 0 (0.0%) | 0 (0.0%) | 0 (0.0%) | N/A |
| Adverse event at 72 hours | 0 (0.0%) | 0 (0.0%) | 0 (0.0%) | 0 (0.0%) | N/A |

Statistics were analyzed by the Chi-square test for categorical data and the one-way ANOVA test for continuous data. Drug adverse events were one diarrhea in the herbal tea group and one dry mouth in the domperidone group. NICU admissions were either from respiratory distress syndrome or transient tachypnea of the newborn. NICU, neonatal intensive care unit; N/A, not applicable; PPH, postpartum hemorrhage; SD, standard deviation.

difference in breast milk volume between Wang Nam Yen herbal tea group at 57.5 ± 50.7 ml and placebo control group at 31.9 ± 27.7 ml (*p* = 0.007) while there was no difference between Wang Nam Yen herbal tea group at 57.5 ± 50.7 ml and domperidone group at 60.9 ± 70.7 ml (*p* = 0.806) (Table 3). While there was a significant difference in baseline breast milk volume at 24 hours postpartum among the three groups (*p* = 0.036), the significant difference between groups was not observed with Bonferroni correction: *p* = 0.039 for herbal tea and control groups, *p* = 0.433 for domperidone and control groups, and *p* = 0.089 for herbal tea and domperidone groups. There was no significant breast milk volume observed among the three groups at 48 hours postpartum (*p* = 0.256). Fig 2 provided the change in total breastmilk volume per expression at each time point with mean and 95% confidence interval.

There were no significant differences in maternal or neonate complications among the three groups (Table 3). For maternal complications, one participant in the control group developed postpartum endometritis. For drug adverse events, one participant in the domperidone group reported dry mouth and one participant in the herbal tea group reported diarrhea as an adverse effect during the first 24 hours postpartum and the symptoms persisted for 72 hours postpartum, subsiding only after the end of the trial. Six infants developed jaundice, two in the control and four in the herbal tea group, and six infants required neonatal intensive care unit admission due to respiratory distress syndrome or transient tachypnea of the newborn: one in Wang Nam Yen herbal group, two in the domperidone group, and three in the placebo group.

Based on participant satisfaction outcomes, 64.2% of all the participants liked their breastfeeding experience during the study period, 71.7% liked using galactagogues to increase their milk production and 75.8% would use galactagogues in their subsequent pregnancy, while

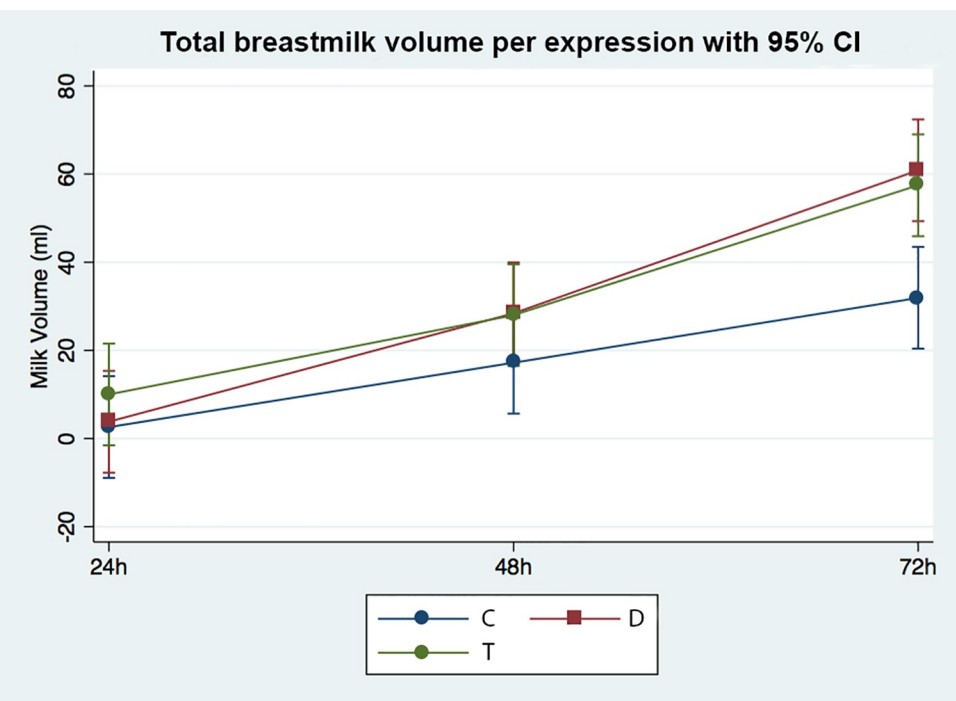

**Fig 2. Change in human milk volume over time.** Mean and 95% confidence intervals are presented. C, control group; D, domperidone group; T, herbal tea group.

18.3% of participants worried about the adverse effects of using galactagogues (Table 4). While not significantly different between the intervention groups, more participants in the control group (32.5%) reported having problems with breastfeeding than participants in the domperidone (20.0%) and herbal tea group (17.5%), and also more participants in herbal tea (62.5%) and domperidone (55.0%) groups agreed to continue the use of galactagogue post-study compared to the control group (35.0%). Of the participating women 68.3% said they would recommend galactagogues to a friend, 75% said they would recommend food to enhance milk production and 78.3% said they would recommend behavior modifications to enhance milk production.

## Discussion

In this study, Wang Nam Yen herbal tea effectively increased breast milk volume production at 72 hours postpartum compared to placebo. There was no significant difference in breast milk volume between Wang Nam Yen herbal tea and the domperidone group. The difference in early augmentation effect at 24 hours postpartum was observed among three groups, but there was no significant difference between any two groups when applying Bonferroni correction. Wang Nam Yen herbal tea and domperidone had no serious side effects on the mothers or infants. One participant reported diarrhea and one participant reported dry mouth as adverse effects from the herbal tea and domperidone group, respectively. The overall satisfaction of using galactagogues was high and most participants would recommend using galactagogues to friends for increasing milk volume production.

Although the study did not measure prolactin levels, the findings demonstrate the efficacy of herbal tea in enhancing milk volume. While the method of collecting milk, using a breast pump to measure milk volume, might affect the study outcome as electronic pumps can

**Table 4. Secondary outcome measures for participant satisfaction.**

| Questions | Total (n = 120) | Herbal tea (n = 40) | Domperidone (n = 40) | Control (n = 40) | p-value |
|---|---|---|---|---|---|
| 1. You like your breastfeeding | | | | | 0.835 |
| Agree | 77 (64.2%) | 28 (70.0%) | 24 (60.0%) | 25 (62.5%) | |
| Disagree | 4 (3.3%) | 1 (2.5%) | 1 (2.5%) | 2 (5.0%) | |
| Not sure | 39 (32.5%) | 11 (27.5%) | 15 (37.5%) | 13 (32.5%) | |
| 2. You like to use this galactagogue | | | | | 0.143 |
| Agree | 86 (71.7%) | 29 (72.5%) | 28 (70.0%) | 29 (72.5%) | |
| Disagree | 3 (2.5%) | 0 (0.0%) | 0 (0.0%) | 3 (7.5%) | |
| Not sure | 31 (25.8%) | 11 (27.5%) | 12 (13.0%) | 8 (20.0%) | |
| 3. You have a problem with your breastfeeding | | | | | 0.200 |
| Agree | 28 (23.3%) | 7 (17.5%) | 8 (20.0%) | 13 (32.5%) | |
| Disagree | 37 (30.8%) | 17 (42.5%) | 10 (25.0%) | 10 (25.0%) | |
| Not sure | 55 (45.8%) | 16 (40.0%) | 22 (55.0%) | 17 (42.5%) | |
| 4. You worried about galactagogues adverse effect | | | | | 0.662 |
| Agree | 22 (18.3%) | 5 (12.5%) | 7 (17.5%) | 10 (25.0%) | |
| Disagree | 56 (46.7%) | 19 (47.5%) | 20 (50.0%) | 17 (42.5%) | |
| Not sure | 42 (35.0%) | 16 (40.0%) | 13 (32.5%) | 13 (32.5%) | |
| 5. You decided to continue using galactagogue | | | | | 0.132 |
| Agree | 61 (50.8%) | 25 (62.5%) | 22 (55.0%) | 14 (35.0%) | |
| Disagree | 10 (8.3%) | 3 (7.5%) | 2 (5.0%) | 5 (12.5%) | |
| Not sure | 49 (40.8%) | 12 (30.0%) | 16 (40.0%) | 21 (52.5%) | |
| 6. You will recommend your friend to use this galactagogue | | | | | 0.171 |
| Agree | 82 (68.3%) | 29 (72.5%) | 27 (67.5%) | 26 (65.0%) | |
| Disagree | 3 (2.5%) | 0 (0.0%) | 0 (0.0%) | 3 (7.5%) | |
| Not sure | 35 (29.2%) | 11 (27.5%) | 13 (32.5%) | 11 (27.5%) | |
| 7. You will recommend your friend to eat food enhancing breast milk production | | | | | 0.791 |
| Agree | 90 (75.0%) | 31 (77.5%) | 28 (70.0%) | 31 (77.5%) | |
| Disagree | 4 (3.3%) | 1 (2.5%) | 1 (2.5%) | 2 (5.0%) | |
| Not sure | 26 (21.7%) | 8 (20.0%) | 11 (27.5%) | 7 (17.5%) | |
| 8. You will advise your friend on behavioral ways to continue breastfeeding | | | | | 0.942 |
| Agree | 94 (78.3%) | 32 (80.0%) | 30 (75.0%) | 32 (80.0%) | |
| Disagree | 5 (4.2%) | 2 (5.0%) | 2 (5.0%) | 1 (2.5%) | |
| Not sure | 21 (17.5%) | 6 (15.0%) | 8 (20.0%) | 7 (17.5%) | |
| 9. You will use galactagogue in your subsequent pregnancy | | | | | 0.412 |
| Agree | 91 (75.8%) | 32 (80.0%) | 30 (75.0%) | 29 (72.5%) | |
| Disagree | 5 (4.2%) | 2 (5.0%) | 0 (0.0%) | 3 (7.5%) | |
| Not sure | 24 (20.0%) | 6 (15.0%) | 10 (25.0%) | 8 (20.0%) | |

Statistics were analyzed by the Chi-square test.

disturb normal biological synthesis of breast milk, the results showed the same trend as previous studies of domperidone at 30 mg/day dose in the efficacy to increase breast milk volume [3, 5, 9, 23, 25]. However, breast milk volume at 72 hours postpartum in the domperidone group showed no significant difference from the placebo control group when using Bonferroni correction. This might occur from the disadvantage of Bonferroni's correction providing smaller-than-required adjusted α which was too conservative, thus often caused false-negative results [26]. While non-breastmilk supplements might affect the amount of breastmilk

production and infant breastfeeding behavior [27], there was no significant difference in formula milk supplementation between the groups in this study. The hospital's standard policy for breastfeeding, including encouraging early breastfeeding, frequent breastfeeding, correct position, and avoiding the use of the formula is important recommendations that should be strictly followed by all mothers before using food, behavior modifications, or galactagogues to increase milk volume production. The findings support the use of galactagogues in addition to behavioral recommendations and food intake to increase breast milk production in mothers with insufficient milk volume following a cesarean delivery [17].

Postpartum endometritis is one of the most common puerperal complications, particularly following cesarean section [28]. While there was a 7% estimated incidence of postpartum endometritis after elective cesarean section [29], no postpartum endometritis in Wang Nam Yen herbal tea group and domperidone group was observed during the trial period. There was no postpartum hemorrhage in Wang Nam Yen herbal tea and the domperidone group. In terms of safety, the side effects of domperidone could include dry mouth, headache, insomnia, abdominal cramps, diarrhea, nausea, and urinary retention [9]. The study found no major side effects, with one report of dry mouth as an adverse effect of domperidone and one report of diarrhea as an adverse effect of the herbal tea, which occurred during the first 24 hours postpartum and continued until the end of the study and after the interventions were stopped the symptoms subsided. Based on the study, there are no safety concerns in using herbal tea during the puerperal period.

There were no significant differences in neonatal outcomes, especially neonatal jaundice, between the intervention and control groups, however, more research may be needed to conclusively support these results [2]. Of 120 infants, six (5%) were admitted to NICU due to respiratory distress syndrome or transient tachypnea of the newborn. The incidence of NICU admission was similar to the previous observational study on term infants of elective cesarean section mothers [30] and slightly lower than term infants of elective repeat cesarean section mothers [31]. No association between the intervention and NICU admission was observed in this study.

As a randomized controlled trial, one of the strengths of this study was by design, as careful randomization and blinding minimized potential confounding factors and biases. This trial was among the first studies to examine the effects of Wang Nam Yen tea compared to domperidone and a placebo. The rationale for factorial design was that domperidone—current medication practice for promoting lactation—is in form of an oral tablet while Wang Nam Yen herbal tea is in form of a teabag needed for infusion of water before use. Thus, the placebo oral tablet and placebo tea were necessary for this study and the two-arm design would not be sufficient. Although the 2x2 design was in our initial research proposal, one arm (herbal tea + domperidone) had to be discarded in response to the comment from the ethics committee. Hence, the potential additive/synergistic effect could not be inferred from our data. Although some participants declined to enroll in the trial before randomization, there were no participants who declined after randomization. Thus, the sample size of this study provided sufficient power and was able to determine the efficacy of Wang Nam Yen herbal tea in increasing breast milk production at 72 hours postpartum as the primary outcome. The results of this study could be applied to pregnant women that underwent cesarean delivery at 37 to 42-week gestational age.

There were several limitations of this study. First, the efficacy of the interventions on other outcomes, such as prolactin level and breast milk volume at 7 days post-administration to confirm hormonal effect and maintenance of augmentation, was not evaluated. Second, while we intended to produce perfect placebo tea with identical external appearance, taste, and aroma. The taste and aroma of placebo tea were difficult to make identical to Wang Nam Yen herbal

tea when placebo tea was needed to produce with the ingredient with no galactagogue effect while providing identical external appearance to Wang Nam Yen herbal tea. This might lead to potential bias when some participants might smell or discuss the aroma and taste with other participants although no explicit action was observed in this study. Third, while this study initially intended to include both preterm and term pregnant women, but they were separated from their child before participation which was the exclusion criteria. Hence, none of the mothers with preterm infants met the eligibility criteria and all included participants were term pregnant (Gestational age 37–42 weeks). Thus, the implication of this study could be applied to only mothers with term pregnancies. Fourth, this study did not record data on whether participants in each group were mainly direct breastfeeding or expressing breastmilk for feeding and did not evaluate lactation insufficiency and sucking problems which might affect the breastmilk production. Moreover, the randomization process was conducted with a single size of block randomization which generally might lead to selection bias if the recruiters knew the block size and previous allocation result, thus the next allocation could be predicted. However, this study tried to minimize the potential selection bias by ensuring that the sealed envelope was opened by a person not involve in the recruitment process and was the only person who knew the allocation. Clinicians doing the recruitment and data collectors were not aware of the allocation. Finally, there was a limitation in sample size calculation due to no previous published peer-reviewed evidence on the effect of this herbal tea. Thus, we calculated the sample size using data based on two parallel groups (domperidone and placebo). This might cause a potential underpowering of the study if the true effect of herbal tea was significantly lower than domperidone. On the other hand, if the true effect of herbal tea was significantly higher than domperidone, only sample size overestimation is concerned.

## Conclusion

Wang Nam Yen herbal tea, an alternative traditional Thai medicine, was effective in augmenting breast milk production at 72 hours postpartum in mothers following cesarean delivery, and there was no evidence that herbal tea and domperidone differed in terms of augmenting breast milk production.

## Supporting information

**S1 Checklist.**
(DOCX)

**S1 File. De-identified data set.**
(XLSX)

**S2 File. Trial study protocol (English).**
(DOCX)

**S3 File. Trial study protocol (Thai).**
(DOCX)

## Acknowledgments

The authors thank Mr. Pinit Chinsoi, Abhaibhubejhr College of Thai Traditional Medicine, and Miss Sawitree Ngamwong, Wang Nam Yen hospital, for preparing the herbal and placebo teas, as well as the Faculty of Pharmaceutical Sciences, Ubon Ratchathani University for preparing the Domperidone. We thank Dr. Charnsak Jungmunkong and contributors from Sunpasitthiprasong Hospital for the administrative support of this trial.

## Author Contributions

**Conceptualization:** Koollachart Saejueng, Krit Pongpirul.

**Data curation:** Koollachart Saejueng, Piyawadee Wuttikonsammakit, Krit Pongpirul.

**Formal analysis:** Koollachart Saejueng, Tanawin Nopsopon.

**Funding acquisition:** Wattanaporn Khumbun, Krit Pongpirul.

**Investigation:** Koollachart Saejueng, Tanawin Nopsopon, Piyawadee Wuttikonsammakit, Wattanaporn Khumbun, Krit Pongpirul.

**Methodology:** Koollachart Saejueng, Tanawin Nopsopon, Krit Pongpirul.

**Project administration:** Koollachart Saejueng, Wattanaporn Khumbun.

**Supervision:** Krit Pongpirul.

**Writing – original draft:** Koollachart Saejueng, Tanawin Nopsopon, Krit Pongpirul.

**Writing – review & editing:** Koollachart Saejueng, Tanawin Nopsopon, Piyawadee Wuttikonsammakit, Wattanaporn Khumbun, Krit Pongpirul.

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
