## [Decision Letter · Decision Letter 0]

10 May 2021

PONE-D-21-05057

Efficacy of Wang Nam Yen herbal tea on human milk production: A randomized controlled trial

PLOS ONE

Dear Dr. Pongpirul,

Thank you for submitting your manuscript to PLOS ONE. After careful consideration, we feel that it has merit but does not fully meet PLOS ONE’s publication criteria as it currently stands. Therefore, we invite you to submit a revised version of the manuscript that addresses the points raised during the review process.

There are a number of issues pointed out by the reviewers. Reviewer 2 has put in a lot of effort and the comments are very useful and if addressed could add a lot of value to the manuscript. Please address all of these either by editing the submission or explaining why you didn't. One of the most pertinent issues is about the onset of action following ingestion of the two galactagogues in the study. Is anything know about how long it would take for the tea to act and at which stage of lactation it has its main effect? More information would be helpful for the reader to understand whether the time from first ingestion to the time of measurement of milk volume is sufficient and also whether a similar effect might be seen if used later in lactation. 

The issue of inclusion of preterm infants is an important one. It would be helpful to give more information about the preterm infants in the study such as the proportion of preterm infants, very preterm infants and full term infants. This would help readers to judge how much impact the inclusion of preterm infants had on the results. Even with the galactagogue the volume of milk expressed is likely to be lower than in a term infant. 

The reviewers were also confused by your use of the terms Day 4 and 3-days. I assume Day 4 means that infant was 3-days old and not yet 4-days old. i.e 72-48 hours of age? I think it would be useful to just use hours of life or a range of hours such as 72-96 hours. 

It would be useful to the reader if you gave more information in the section on statistical analysis about the use of the Bonferroni correction. With the correction in place what p-value was assumed significant? 

We look forward to receiving your revised manuscript.

Kind regards,

Jacqueline J. Ho, MB.ChB, MMedSc(ClinEpid), FRCP, FRCPCH, FRCPI

Academic Editor

PLOS ONE

Journal Requirements:

2. In order to meet our requirements for data accessibility, please provide the following information about the Wang Nam Yen herbal tea prepared by the hospital and the control herbal tea :

a) the sources of all ingredients, including the full name of the company or local market

b) the lot numbers of the ingredients

c) the amount of each ingredient contained in one dose of tea

3. Please provide additional details regarding participant consent. In the ethics statement in the Methods and online submission information, please ensure that you have specified whether you obtained written informed consent from parents or guardians of 15-17 year old participants.

Reviewers' comments:

Reviewer's Responses to Questions

**Comments to the Author**

1. Is the manuscript technically sound, and do the data support the conclusions?

Reviewer #1: Partly

Reviewer #2: Partly

Reviewer #3: Partly

2. Has the statistical analysis been performed appropriately and rigorously? 

Reviewer #1: I Don't Know

Reviewer #2: I Don't Know

Reviewer #3: Yes

3. Have the authors made all data underlying the findings in their manuscript fully available?

Reviewer #1: No

Reviewer #2: No

Reviewer #3: Yes

4. Is the manuscript presented in an intelligible fashion and written in standard English?

Reviewer #1: Yes

Reviewer #2: No

Reviewer #3: Yes

5. Review Comments to the Author

Reviewer #1: This paper reports on a randomised trial of two approaches to breast milk production in women undergoing Caesarean delivery.

There area number of issues with this paper

The design is a three arm trial -the authors need to explain why a 2x2 design was not looked at as this would help see whether the effects of the interventions were additive.

The sample size is not given in sufficient detail to allow someone to reproduce it - especially as the primary comparison would be a 3-way one and the actual effect sizes and sd are not given in the paper.

The allocation method was via sealed envelopes presumably containing the allocation - the investigator is therefore not blind. A block size of three (presumably one per arm per block) gives the opportunity of significant treatment foreknowledge and as such the integrity of the randomisation process is not guaranteed. What tests were performed to ensure envelopes were opened in order, and were those consenting women kept in ignorance of all previous treatment allocations? What were the reasons for not entering and did they differ by next allocation?

72 hours is 3 days post delivery - why are we variously told about 4 day outcomes?

It is statistically incorrect to test at baseline (Table I).

What methods are used when? ANOVA requires iid Normal residuals, Kruskal-Wallis would be used when this isn't the case. Why are they being performed simultaneously? Please explain the tests used at each point and give contrasts and CI wherever possible.

Drug AEs are not assessed similarly between groups and need to be removed or assessed blind to allocation.

Reviewer #2: This study is of good quality with appropriate controls and the authors have followed the guidelines recommended by CONSORT for RCT-studies. The authors have explained that all data relevant to the study have been included in the article and any de-identified data could be shared upon request. The text is quite well written. However, most of the text is written in past tense, and in certain areas, this can be misleading. An example is the first sentence in the introduction section. Perhaps the authors can re-check the whole manuscript to ensure that the tense used is appropriate.

Further clarifications are needed in these sections:

Introduction:

i. To justify the measurement of the outcome at 72 hours, I suggest that the authors provide data for both Wang Nam Yen and domperidone (with appropriate reference to the literature) on how long after ingestion will a galactogenic effect be expected.

ii. A reference is needed for the last few sentences in the 2nd paragraph, and the first sentence of the 3rd paragraph of this section.

iii. Do the authors have data on which component of the Wang Nam Yen is the most active galactagogue? Some discussion on this would be interesting.

Study design:

i. It is not very clear when the participants were recruited. Was it before or after delivery?

ii. Newborn-mother separation is one of the exclusion criteria. Please clarify why were the 6 mothers whose babies were admitted to NICU (thus meeting the exclusion criteria) included in the study.

Participants:

i. Premature infants were included. Their sucking reflex and chances for direct breastfeeding could be impaired. The main stimulant for more breast milk production is direct breastfeeding and expression of breast milk which may not go well with premature infants. How did the authors manage this?

ii. It would be important for readers to know the rationale behind why only mothers who had undergone caesarean section was included in this study.

Randomization and blinding:

i. The authors have taken trouble to ensure each group received similar looking herbal tea bags and tablets. However, it would be important to mention if the intervention tea and the placebo tea will both have the same appearance, taste, and smell. The same goes for the tablets (domperidone vs placebo).

ii. It would be important to mention what was actually used to fill in the placebo tea bags and what was the ingredient used in the placebo tablet, so that readers can be certain that these placebos do not have any galactogenic effect.

iii. Details of how the random sequence was generated and how the allocation concealment was done (which was nicely written in the protocol) should be added into the manuscript.

iv. Other important details available from the protocol such as the availability of similar breastfeeding support to all the groups should also be included into the manuscript.

Intervention:

i. The sentence "Domperidone was administered at 10 mg per dose, as 30 mg of domperidone per day" is rather confusing. Could this be written in as “Domperidone was given 10mg 3 times per day (total dose 30mg daily)."?

Outcomes measures:

i. Clarification is needed on what was the unit of analysis used for the milk volume measurements.

Was the recorded milk volume a single measurement of 'the total amount' obtained from both breasts after a single expression (mls/extraction) OR was it an accumulation of milk volume measured several times a day (mls/day)?

ii. I note that the milk expression is done 2 hours after the baby has last fed. Can the authors clarify how milk collection was done for babies that demand hourly feeding?

iii. Did the mother express her milk herself or did a trained staff help her to do it? This is to see how uniform the milk expression was done.

iv. It would be important to describe how the babies were mainly fed in each group.

• Were the mothers mainly directly breastfeeding or were many of them expressing milk for feeding? It would be useful to know this proportion because frequent effective direct breastfeeding is a powerful stimulator for breast milk production and this could affect the end result.

v. Since supplementing feeding with formula can affect the amount of breastmilk produced, it would be important to mention how many babies were supplemented with milk in each group.

vi. Formula milk supplementation might also affect the baby’s intake and cause less breast emptying, thus a false sense of ‘more breastmilk’. The authors could elaborate on how this is managed.

vii. Table 1 has lots of details such as method of payment, number of antenatal visits and presence of labour symptoms. Perhaps the authors could explain why these information were included. If they do not contribute to breast milk production i.e. potential confounders, perhaps they can omit these details so that the table is less cluttered.

viii. There is a statement "There were no significant differences between the groups in …….72 hours postpartum period or the amount of formula milk required (Table 1)." The amount of formula milk required was not presented in Table 1. I hope the authors will include this in the table.

Results:

i. Mothers with premature babies may have different milk volume production because of physiological reasons and difference in infant sucking abilities. Therefore, the number of mothers with premature babies in each group should be mentioned.

ii. Being able to directly breastfeed a baby can make a difference in milk supply. It would be useful for the authors to provide data (if available) on the proportions of mothers directly breastfeeding in each group.

iii. Do the authors have data on whether the participants have lactation insufficiency, and if so, how many were there in each group? It would be useful for this to be mentioned because lactation insufficient women may produce milk differently from lactation sufficient women. The author’s definition of what they consider lactation insufficient will also be needed if this data is included.

iv. It would be better and clearer if the actual volume and SD (baseline and at 72 hours) were reported. If possible, display in a table. Showing odds ratio of linear regression (95% confidence interval) could be more meaningful than just the p value. Readers will be interested to know the values obtained to support the basis that taking Wang Nam Yen or Domperidone would make a meaningful difference.

v. In addition, the results described narratively in the text and the results displayed in the graph did not seem to match and were confusing. The text stated that the milk volume in Group T was significantly higher than that in Group C, but in the graph, both Group T and Group D were plotted very closed to each other at 72 hours. On the other hand, the text stated that milk volume for Group D and C were not statistically different, but in the graph, they were obviously plotted very far from each other. Further clarification is needed.

vi. The unit of analysis for Figure 2 is needed in the title of the figure. Was it a mean volume of milk per day at 72 hours of life or volume of milk per expression at 72 hours of life, or was it a mean increment of milk volume since baseline?

vii. The authors did mention that milk volume in the domperidone group showed no significant difference from the placebo group after applying Bonferroni correction.

• Please clarify what was meant by the phrase 'too conservative'.

• What confounders was used?

viii. About the adverse effects and Table 2:

• Why was postpartum endometritis highlighted separately in Table 2? If this could be related to the intervention, perhaps it should be discussed in the discussion section as well?

• I am confused with the actual number of adverse events. The text mentioned that 3 mothers suffered adverse events and only mentioned 3 types of adverse events, namely postpartum endometriosis, dry mouth and diarrhoea. However, Table 2 reported 6 types of adverse events (headache, dry mouth, diarrhea, muscle cramps, itching, or allergic reactions) and that there was a total of 6 mothers experiencing them. Could the authors clarify?

• The authors took the trouble to separate out adverse effects occurring at 24h, 48h and 72h in Table 2. Is there a reason to do so? If there is, perhaps details should then be provided as to which adverse effect occurred at which time, and discuss this in the discussion section as well?

ix. About the 6 infants that required NICU admission:

• What was the reason behind the 6 infants requiring ICU admission?

• How were these mother-infant pairs handled when separation did happen after randomization?

• How many were there in each group?

Clear explanation and details would allow readers to know if the admissions were related to intervention and if intention to treat analysis was performed.

x. The manuscript reported some participants experiencing adverse effects which subsided after discontinuing the intervention. Further details are needed to see how were they related to the intervention or could affect the analysis.

• How many other participants in each group did not actually complete the intervention as prescribed, ie did not take it 3 times a day for 3 days?

• At which time point did these participants in each group stopped taking the intervention?

• Was the milk expression continued at each time point for analysis? If yes, was the participants analysed based on their randomized group?

Perhaps these details could be included in the flow diagram or at least in the text of the results.

Discussion

i. The manuscript stated "Although some participants declined to enroll in the trial following randomization, …..". This line needs clarification because it contradicts the flow diagram that showed that participants who declined to enroll were not randomised. If these were participants who had originally consented to take part in the study, but pulled out after randomization, they should be reflected in the study flow diagram as participant drop-outs, hence the final number of participants who were analysed would be less than 40 in each group. The numbers in Table 2 and 3 will also need adjustments. The reasons for dropping out (if any) should be described.

ii. Data on confounding factors to milk production should be stated. This includes:

• when breastfeeding was first initiated (the protocol says that mothers were encouraged to breastfeed within 24 hours, but having her first breastfeed at birth or first breastfeed at 18 hours of life is a big confounder)

• babies were supplemented with formula in each group

• number of infants with sucking problems

• number of mothers with lactation insufficiency

. If they were not available, they could be discussed as limitations of the study.

Reviewer #3: This is an interesting study that seeks to compare the effectiveness of Wang Nam Yen herbal tea vs. domperidone compared to placebo in achieving improved mother's milk production at 72 hrs following delivery. There were 120 participants in the trial who ranged in delivery gestational age between 28-40 weeks, with a mean GA of ~38 wks and a bw of ~3 kg. The treatment was prescribed for 72 hrs with milk volume measurements via pumped milk expression at each 24 hr interval. There were statistically significant differences between Wang Nam Yen herbal tea and placebo, but not between domperidone and placebo or between Wang Nam Yen herbal tea and domperidone. These are important findings that will serve as the basis for future studies. The following are points that should be addressed to strengthen the clarity and strength of the findings:

(1) In the Methods, the placebo tea needs to be delineated and described. How were the participants and investigators blinded given that herbal teas with certain ingredients have particular aromas?

(2) In the Methods, please describe the block size of randomization.

(3) Methods: in the sample size estimate, you mention milk volume at 4 days but this analysis was at 72 hrs. Perhaps you are using data for power that described milk volume differences at 4 days for your sample size calculation. Please rewrite this section.

(4) Methods: it is concerning that women who delivered preterm were analyzed with women who deliver term. I think this is a potential weakness of the study design. Further, in the analyses of neonatal side-effects, the study is not powered to look at preterm neonatal outcomes by treatment. The tea and domperidone may not differ in their safety profile with preterm mothers, but you would need far more mothers enrolled in such a study. I do not think you can extrapolate safety to those born <35wks so easily. You might suggest that this be refined in future studies.

(5) The Discussion needs to be checked for accuracy for scientific English to make it more readable. There are grammatical errors in that section that are not seen in prior sections.

6. PLOS authors have the option to publish the peer review history of their article (what does this mean?). If published, this will include your full peer review and any attached files.

Reviewer #1: No

Reviewer #2: **Yes: **Wai Cheng Foong

Reviewer #3: No

---

## [Author Response · Author response to Decision Letter 0]

22 May 2021

Dear Editor,

We thank you and the reviewers for the comments and suggestions. We have tried our best to address all of the points raised by you and the reviewers during the past 12 days offered. Please find our point-by-point responses below:

Editor: Thank you for submitting your manuscript to PLOS ONE. After careful consideration, we feel that it has merit but does not fully meet PLOS ONE’s publication criteria as it currently stands. Therefore, we invite you to submit a revised version of the manuscript that addresses the points raised during the review process.

Response: Thank you for your time and consideration. The manuscript was revised to meet PLOS ONE’s publication criteria.

Editor: There are a number of issues pointed out by the reviewers. Reviewer 2 has put in a lot of effort and the comments are very useful and if addressed could add a lot of value to the manuscript. Please address all of these either by editing the submission or explaining why you didn't. One of the most pertinent issues is about the onset of action following ingestion of the two galactagogues in the study. Is anything know about how long it would take for the tea to act and at which stage of lactation it has its main effect? More information would be helpful for the reader to understand whether the time from first ingestion to the time of measurement of milk volume is sufficient and also whether a similar effect might be seen if used later in lactation.

Response: Thank you for your comment. We greatly appreciate Reviewer 2 for the useful pieces of advice. The information about the onset of action following ingesting of the two galactagogues was added in the Introduction section.

Editor: The issue of inclusion of preterm infants is an important one. It would be helpful to give more information about the preterm infants in the study such as the proportion of preterm infants, very preterm infants and full term infants. This would help readers to judge how much impact the inclusion of preterm infants had on the results. Even with the galactagogue the volume of milk expressed is likely to be lower than in a term infant.

Response: We are sorry for the confusing inclusion criteria and discussion. Initially, we intended to include both preterm and term pregnant women, but they were separated from their child before participation which was one of the exclusion criteria. Hence, all included participants were term pregnant (Gestational age 37–42 weeks). To minimize confusion, the inclusion criteria were amended from the first registered protocol. The inclusion criteria in the Methods section were revised and the reason for inclusion criteria amendment was provided in the Discussion section.

Editor: The reviewers were also confused by your use of the terms Day 4 and 3-days. I assume Day 4 means that infant was 3-days old and not yet 4-days old. i.e 72-48 hours of age? I think it would be useful to just use hours of life or a range of hours such as 72-96 hours. 

Response: Thank you for pointing out and sorry for the confusion. The terms were revised for clarification throughout the manuscript.

Editor: It would be useful to the reader if you gave more information in the section on statistical analysis about the use of the Bonferroni correction. With the correction in place what p-value was assumed significant? 

Response: An additional information on Bonferroni correction was added in the Methods section. P-value was assumed significant at p < 0.017.

Editor: 1. Please ensure that your manuscript meets PLOS ONE's style requirements, including those for file naming.

Response: The manuscript was revised to meet PLOS ONE’s style requirements.

Editor: 2. In order to meet our requirements for data accessibility, please provide the following information about the Wang Nam Yen herbal tea prepared by the hospital and the control herbal tea: a) the sources of all ingredients, including the full name of the company or local market; b) the lot numbers of the ingredients; c) the amount of each ingredient contained in one dose of tea

Response: All ingredients for the Wang Nam Yen herbal tea and the control herbal tea were from Charoensuk 

Pharma Supply Co.,Ltd. (https://www.cpspharma.co.th/). Unfortunately, no lot number was provided by the company. Each Wang Nam Yen herbal tea bag contained 500 mg sappan (Caesalpinia sappan Linn.), 500 mg licorice (Glycyrrhiza glabra Linn.), 500 mg bale fruit (Aegle marmelos L. Corr), 500 mg ginger (Zingiber officinale Roscoe), and 500 mg jewel vine (Derris scandens (Roxb.) Benth). The herbal tea placebo was also prepared as a teabag for each meal, and the placebo tea bag was identical to Wang Nam Yen teabag. The placebo tea bag was filled with 2,500 mg pandan (Pandanus amaryllifoilus) leaves. The information was provided in the Methods section.

Editor: 3. Please provide additional details regarding participant consent. In the ethics statement in the Methods and online submission information, please ensure that you have specified whether you obtained written informed consent from parents or guardians of 15-17 year old participants.

Response: Additional details on participant consent were added in the Methods and online submission information.

Editor: 4. We note that you have indicated that data from this study are available upon request. PLOS only allows data to be available upon request if there are legal or ethical restrictions on sharing data publicly. For information on unacceptable data access restrictions, please see http://journals.plos.org/plosone/s/data-availability#loc-unacceptable-data-access-restrictions.

Response: There are no restrictions. De-identified data set was provided as a Supporting Information file.

Reviewer #1: This paper reports on a randomised trial of two approaches to breast milk production in women undergoing Caesarean delivery. There are a number of issues with this paper. The design is a three arm trial -the authors need to explain why a 2x2 design was not looked at as this would help see whether the effects of the interventions were additive.

Response: Thank you for pointing this out. The rationale for factorial design was that domperidone—the current standard medication for promoting lactation—is in form of an oral tablet while Wang Nam Yen herbal tea is in form of a teabag needed for infusion of water before use. Thus, the placebo oral tablet and placebo tea were necessary for this study. The 2x2 design was in our initial research proposal but one arm (herbal tea + domperidone) had to be discarded in response to the comment from the ethics committee. Hence, the potential additive/synergistic effect could not be inferred from our data.

Reviewer #1: The sample size is not given in sufficient detail to allow someone to reproduce it - especially as the primary comparison would be a 3-way one and the actual effect sizes and sd are not given in the paper.

Response: Thank you for your comment. The sample size calculation subheading was revised in the Method section to provide more reproducibility of the study.

Reviewer #1: The allocation method was via sealed envelopes presumably containing the allocation - the investigator is therefore not blind. A block size of three (presumably one per arm per block) gives the opportunity of significant treatment foreknowledge and as such the integrity of the randomisation process is not guaranteed. What tests were performed to ensure envelopes were opened in order and were those consenting women kept in ignorance of all previous treatment allocations? What were the reasons for not entering and did they differ by next allocation? 

Response: Thank you for pointing this out. The randomization and blinding subheadings were revised in the Methods section to clarify the process. The allocation method was deliberately ensured that the investigator was blinded. That is, after generating randomization codes, the research team responsible for blinding created an opaque envelope to seal the randomization codes. The front end of an opaque sealed envelope presented the study name, principal investigator name, site of study, randomization ID, study ID, and date which envelope will be opened. There was a document inside the sealed envelope with randomization ID, assigned group, study ID, date and time which envelope will be opened, and signature of envelope opener who are not clinician nor data collector. While a single block size (of three) might cause potential treatment foreknowledge and lead to selection bias, in theory, this trial random allocation was done by a separate research team independent from enrollment personnel, clinician, or data collector to prevent the selection bias.

Reviewer #1: 72 hours is 3 days post delivery - why are we variously told about 4 day outcomes?

Response: We are sorry for the confusion. The terms were revised for clarification throughout the manuscript.

Reviewer #1: It is statistically incorrect to test at baseline (Table I).

Response: Thank you for your comment. The p-values presented in Table 1 were to demonstrate the comparable baseline characteristics of participants across the three groups after the randomization. We believe this statistical information is useful but could be removed as appropriate.

Reviewer #1: What methods are used when? ANOVA requires iid Normal residuals, Kruskal-Wallis would be used when this isn't the case. Why are they being performed simultaneously? Please explain the tests used at each point and give contrasts and CI wherever possible.

Response: Thank you for pointing out a concerning issue. ANOVA and Kruskal-Wallis were not performed simultaneously. ANOVA was used when the continuous variable was normally distributed while Kruskal-Wallis was used when the continuous variable was not normally distributed. The statistical analysis subheading in the Methods section was revised for clarification. Footnote on which statistical test was performed for each variable was added in Table 1.

 

Reviewer #1: Drug AEs are not assessed similarly between groups and need to be removed or assessed blind to allocation.

Response: The allocation concealment was deliberately performed to ensure double-blinded as stated above. Drug AEs were assessed blindly and similar among groups. The Methods section was revised to provide more detail and clarification on randomization and blinding, and outcome measurement.

Reviewer #2: This study is of good quality with appropriate controls and the authors have followed the guidelines recommended by CONSORT for RCT-studies. The authors have explained that all data relevant to the study have been included in the article and any de-identified data could be shared upon request. The text is quite well written. However, most of the text is written in past tense, and in certain areas, this can be misleading. An example is the first sentence in the introduction section. Perhaps the authors can re-check the whole manuscript to ensure that the tense used is appropriate.

Response: Thank you for the compliments. Appropriate use of tense was revised throughout the manuscript.

Reviewer #2: Introduction: i. To justify the measurement of the outcome at 72 hours, I suggest that the authors provide data for both Wang Nam Yen and domperidone (with appropriate reference to the literature) on how long after ingestion will a galactogenic effect be expected.

Response: Thank you for your comment. Additional details on the expected onset of lactogenic effect for both Wang Nam Yen and domperidone were added in the Introduction section.

Reviewer #2: Introduction: ii. A reference is needed for the last few sentences in the 2nd paragraph, and the first sentence of the 3rd paragraph of this section.

Response: Thank you for your comment. The references were added as suggested accordingly.

Reviewer #2: Introduction: iii. Do the authors have data on which component of the Wang Nam Yen is the most active galactagogue? Some discussion on this would be interesting.

Response: Thank you for your comment. Additional information on the most active galactagogue was added in the Introduction section.

Reviewer #2: Study design: i. It is not very clear when the participants were recruited. Was it before or after delivery?

Response: The participants were recruited after delivery by cesarean section. Additional detail was added in the Method section for clarification.

Reviewer #2: Study design: ii. Newborn-mother separation is one of the exclusion criteria. Please clarify why were the 6 mothers whose babies were admitted to NICU (thus meeting the exclusion criteria) included in the study.

Response: Study design: 6 babies were admitted to NICU after participation but did not separate from their mothers whereas the exclusion criteria excluded newborn-mother separation before participation. The Method section was revised to clarify the eligibility criteria.

Reviewer #2: Participants: i. Premature infants were included. Their sucking reflex and chances for direct breastfeeding could be impaired. The main stimulant for more breast milk production is direct breastfeeding and expression of breast milk which may not go well with premature infants. How did the authors manage this?

Response: We are sorry for the confusing inclusion criteria and discussion. Initially, we intended to include both preterm and term pregnant women, but they were separated from their child before participation which was one of the exclusion criteria. Hence, all included participants were term pregnant (Gestational age 37-42 weeks). To minimize confusion, the inclusion criteria were amended from the first registered protocol. The inclusion criteria in the Methods section were revised and the reason for inclusion criteria amendment was provided in the Discussion section.

Reviewer #2: ii. It would be important for readers to know the rationale behind why only mothers who had undergone caesarean section was included in this study.

Response: Women that have undergone cesarean deliveries were selected as the population in this study due to cesarean delivery was a risk factor of inadequate breast milk production. Additional detail was provided in the Introduction section.

Reviewer #2: Randomization and blinding: i. The authors have taken trouble to ensure each group received similar looking herbal tea bags and tablets. However, it would be important to mention if the intervention tea and the placebo tea will both have the same appearance, taste, and smell. The same goes for the tablets (domperidone vs placebo).

Response: Thank you for pointing it out. Additional detail on placebo tea and tablet was provided in the Methods section. While we intended to produce an ideal placebo tea, the taste and aroma of placebo tea were still different from the Wang Nam Yen herbal tea. However, no participants had a chance to take both active and placebo tea simultaneously. An additional discussion on this limitation was provided in the Discussion section.

Reviewer #2: ii. It would be important to mention what was actually used to fill in the placebo tea bags and what was the ingredient used in the placebo tablet, so that readers can be certain that these placebos do not have any galactogenic effect.

Response: The information of ingredients used in placebo tea bags and tablets were added in the Method section accordingly.

Reviewer #2: iii. Details of how the random sequence was generated and how the allocation concealment was done (which was nicely written in the protocol) should be added into the manuscript.

Response: The details of randomization and allocation concealment were added to the Methods section accordingly.

Reviewer #2: iv. Other important details available from the protocol such as the availability of similar breastfeeding support to all the groups should also be included into the manuscript.

Response: The Methods section was revised as suggested accordingly.

Reviewer #2: Intervention: i. The sentence "Domperidone was administered at 10 mg per dose, as 30 mg of domperidone per day" is rather confusing. Could this be written in as “Domperidone was given 10mg 3 times per day (total dose 30mg daily)."?

Response: The sentence was revised as suggested accordingly.

Reviewer #2: Outcomes measures: i. Clarification is needed on what was the unit of analysis used for the milk volume measurements. Was the recorded milk volume a single measurement of 'the total amount' obtained from both breasts after a single expression (mls/extraction) OR was it an accumulation of milk volume measured several times a day (mls/day)?

Response: Thank you for your comment. The unit of analysis was the total amount of milk volume in a single expression (mL/extraction). The Methods section was revised accordingly.

Reviewer #2: Outcomes measures: ii. I note that the milk expression is done 2 hours after the baby has last fed. Can the authors clarify how milk collection was done for babies that demand hourly feeding?

Response: The milk collection was done uniformly for all participants. While some babies might demand hourly feeding, all mothers were notified at 2 hours before each milk collection section which allowed the last breastfeeding prior to the milk collection. The Methods section was revised accordingly.

Reviewer #2: Outcomes measures: iii. Did the mother express her milk herself or did a trained staff help her to do it? This is to see how uniform the milk expression was done.

Response: A trained nurse helped all participants to express their milk. Additional detail was added to the Methods section accordingly.

 

Reviewer #2: Outcomes measures: iv. It would be important to describe how the babies were mainly fed in each group. Were the mothers mainly directly breastfeeding or were many of them expressing milk for feeding? It would be useful to know this proportion because frequent effective direct breastfeeding is a powerful stimulator for breast milk production and this could affect the end result.

Response: Thank you for your comment. Unfortunately, there was no recorded data on the proportion of mothers directly breastfeeding in each group. This was one of the limitation of the study and the Discussion section was revised accordingly.

Reviewer #2: Outcomes measures: v. Since supplementing feeding with formula can affect the amount of breastmilk produced, it would be important to mention how many babies were supplemented with milk in each group.

Response: Thank you for your comment. Information on formula milk supplement measure was added in the Methods and Results section.

Reviewer #2: Outcomes measures: vi. Formula milk supplementation might also affect the baby’s intake and cause less breast emptying, thus a false sense of ‘more breastmilk’. The authors could elaborate on how this is managed.

Response: The need for formula milk supplementation was recorded daily and categorized into 4 groups: no formula milk supplement, required 1-day supplement, required 2 days supplement, and required 3 days supplement. Association between intervention groups and formula milk supplementation was evaluated by Chi-square test. Additional details were added in the Result and the Discussion sections.

Reviewer #2: Outcomes measures: vii. Table 1 has lots of details such as method of payment, number of antenatal visits and presence of labour symptoms. Perhaps the authors could explain why these information were included. If they do not contribute to breast milk production i.e. potential confounders, perhaps they can omit these details so that the table is less cluttered.

Response: Most variables could be potential confounders of breast milk production while some variables were potential confounders for a secondary outcome such as the method of payment, presence of labor symptoms, low socioeconomics status could be potential confounders of postpartum endometriosis. Variable without evidence for potential confounders were omitted from Table 1 as advised.

Reviewer #2: Outcomes measures: viii. There is a statement "There were no significant differences between the groups in …….72 hours postpartum period or the amount of formula milk required (Table 1)." The amount of formula milk required was not presented in Table 1. I hope the authors will include this in the table.

Response: We apologize for the missing information in Table 1. The need for formula milk supplementation was recorded daily and categorized into 4 groups: no formula milk supplement, required 1-day supplement, required 2 days supplement, and required 3 days supplement. The missing information was added in Table 1 accordingly.

Reviewer #2: Results: i. Mothers with premature babies may have different milk volume production because of physiological reasons and difference in infant sucking abilities. Therefore, the number of mothers with premature babies in each group should be mentioned.

Response: As mentioned above, all included participants were term pregnant; no premature infants were included.

Reviewer #2: Results: ii. Being able to directly breastfeed a baby can make a difference in milk supply. It would be useful for the authors to provide data (if available) on the proportions of mothers directly breastfeeding in each group.

Response: Thank you for your comment. Unfortunately, there was no recorded data on the proportion of mothers directly breastfeeding in each group. This was one of the limitation of the study and the Discussion section was revised accordingly.

Reviewer #2: Results: iii. Do the authors have data on whether the participants have lactation insufficiency, and if so, how many were there in each group? It would be useful for this to be mentioned because lactation insufficient women may produce milk differently from lactation sufficient women. The author’s definition of what they consider lactation insufficient will also be needed if this data is included.

Response: Thank you for your comment. We did not evaluate lactation insufficiency in this study. I agree that lactation insufficiency might affect breast milk production. This was a limitation of the study and additional discussion was added in the Discussion section.

Reviewer #2: Results: iv. It would be better and clearer if the actual volume and SD (baseline and at 72 hours) were reported. If possible, display in a table. Showing odds ratio of linear regression (95% confidence interval) could be more meaningful than just the p value. Readers will be interested to know the values obtained to support the basis that taking Wang Nam Yen or Domperidone would make a meaningful difference.

Response: Thank you for your comment. The actual volume and SD of breast milk were added in Table 2.

Reviewer #2: Results: v. In addition, the results described narratively in the text and the results displayed in the graph did not seem to match and were confusing. The text stated that the milk volume in Group T was significantly higher than that in Group C, but in the graph, both Group T and Group D were plotted very closed to each other at 72 hours. On the other hand, the text stated that milk volume for Group D and C were not statistically different, but in the graph, they were obviously plotted very far from each other. Further clarification is needed.

Response: While the statistical test compared mean and SD, the graph plot presented mean with a 95% confidence interval. The distance between the groups in the graph presented a close distance between Group T and Group D while a far distance of Group T or Group D from Group C. The graph plot reflected statistically significant unadjusted results between Group T and Group C, Group D, and Group C. However, when Bonferroni correction was applied, the adjusted result was more valid and might differ from the graph plot.

Reviewer #2: Results: vi. The unit of analysis for Figure 2 is needed in the title of the figure. Was it a mean volume of milk per day at 72 hours of life or volume of milk per expression at 72 hours of life, or was it a mean increment of milk volume since baseline?

Response: Thank you for your comment. The unit of analysis (volume of milk per expression) was added as suggested accordingly.

Reviewer #2: Results: vii. The authors did mention that milk volume in the domperidone group showed no significant difference from the placebo group after applying Bonferroni correction. Please clarify what was meant by the phrase 'too conservative'. What confounders was used?

Response: An additional detail on Bonferroni correction was added in the Methods and Discussion sections. The disadvantage of Bonferroni's correction was small-than-required adjusted α, thus often caused false-negative result. ‘Too conservative’ is equivalent to ‘weak statistical power’. Potential confounders in this study were maternal age, socioeconomic status, method of payment, gestational age, parity status, adequate antenatal care, presence of labor symptom, previous cesarean section, the experience of breastfeeding, neonatal birth weight, postpartum hemorrhage, fluid balance, and formula milk required.

Reviewer #2: Results: viii. About the adverse effects and Table 2: Why was postpartum endometritis highlighted separately in Table 2? If this could be related to the intervention, perhaps it should be discussed in the discussion section as well?

Response: We highlighted postpartum endometritis because this complication was common in cesarean delivery. Information that Wang Nam Yen herbal tea will not significantly increase postpartum endometritis compared to domperidone and placebo could provide more clear safety profile of Wang Nam Yen herbal tea. An additional discussion was added in the Discussion section.

Reviewer #2: Results: I am confused with the actual number of adverse events. The text mentioned that 3 mothers suffered adverse events and only mentioned 3 types of adverse events, namely postpartum endometriosis, dry mouth and diarrhoea. However, Table 2 reported 6 types of adverse events (headache, dry mouth, diarrhea, muscle cramps, itching, or allergic reactions) and that there was a total of 6 mothers experiencing them. Could the authors clarify?

Response: We apologize for the confusing information. There were 3 mothers who suffered adverse events: one with postpartum endometriosis in the control group (we displayed this event in Maternal complication), one with dry mouth in the domperidone group, one with diarrhea in Wang Nam Yen herbal tea group. Table 2 legend “Drug adverse event includes headache, dry mouth, diarrhea, muscle cramps, itching, or allergic reactions.” was intended to mention which symptoms could be considered as drug adverse event, not the actual drug adverse events occurred. 

Thus, Table 2 legend was revised to present actual drug adverse events only.

Reviewer #2: Results: The authors took the trouble to separate out adverse effects occurring at 24h, 48h and 72h in Table 2. Is there a reason to do so? If there is, perhaps details should then be provided as to which adverse effect occurred at which time, and discuss this in the discussion section as well?

Response: We recorded adverse events daily for two reasons. First, there was a lack of evidence on when the Wang Nam Yen herbal tea might cause adverse events. Second, we intended to assess how long the adverse events persisted and subsided. Additional details were added to the Results and the Discussion sections.

Reviewer #2: Results: ix. About the 6 infants that required NICU admission: What was the reason behind the 6 infants requiring ICU admission? How were these mother-infant pairs handled when separation did happen after randomization? How many were there in each group? Clear explanation and details would allow readers to know if the admissions were related to intervention and if intention to treat analysis was performed.

Response: The six infants were admitted to NICU due to respiratory distress syndrome or transient tachypnea of the newborn (TTNB) which the authors assumed no association with intervention. When mother-infant separation occurred after randomization, the mothers continued in the trial similar to other participants except direct breastfeeding was not possible while the infants received standard of care. There was 1 infant admitted to NICU in Wang Nam Yen herbal tea group, 2 in the domperidone group, and 3 in the placebo group. Additional details were added in the Methods, Results, and Discussion sections.

Reviewer #2: Results: x. The manuscript reported some participants experiencing adverse effects which subsided after discontinuing the intervention. Further details are needed to see how were they related to the intervention or could affect the analysis. How many other participants in each group did not actually complete the intervention as prescribed, ie did not take it 3 times a day for 3 days? At which time point did these participants in each group stopped taking the intervention? Was the milk expression continued at each time point for analysis? If yes, was the participants analysed based on their randomized group? Perhaps these details could be included in the flow diagram or at least in the text of the results.

Response: We apologize for the confusing information. The phrase “which subsided after discontinuing the intervention” means that the adverse symptoms persisted for 72 hours postpartum and then subsided after the trial end (no intervention received). Thus, all participants were complete the intervention as prescribed and analyzed based on their randomization group. The discussion on adverse events was revised in the Discussion section.

Reviewer #2: Discussion: i. The manuscript stated "Although some participants declined to enroll in the trial following randomization, …..". This line needs clarification because it contradicts the flow diagram that showed that participants who declined to enroll were not randomised. If these were participants who had originally consented to take part in the study, but pulled out after randomization, they should be reflected in the study flow diagram as participant drop-outs, hence the final number of participants who were analysed would be less than 40 in each group. The numbers in Table 2 and 3 will also need adjustments. The reasons for dropping out (if any) should be described.

Response: We apologize for the confusing information. While there were some participants who decline to enroll in the trial before randomization, there were no participants who declined after randomization. Relevant detail in the Discussion section was revised accordingly.

Reviewer #2: Discussion: ii. Data on confounding factors to milk production should be stated. This includes: when breastfeeding was first initiated (the protocol says that mothers were encouraged to breastfeed within 24 hours, but having her first breastfeed at birth or first breastfeed at 18 hours of life is a big confounder), babies were supplemented with formula in each group, number of infants with sucking problems, number of mothers with lactation insufficiency. If they were not available, they could be discussed as limitations of the study.

Response: We apologize for the confusing information. The first breastfeeding was first initiated after the first dose of intervention which occurred 12-18 hours following delivery. Encouragement to breastfeed their infant within 24 hours of delivery was Sanpasitthiprasong hospital's standard of care policy. The Methods section was revised accordingly. Need for formula milk supplementation was recorded daily and categorized into 4 groups: no formula milk supplement, required 1-day supplement, required 2 days supplement, and required 3 days supplement. Association between intervention groups and formula milk supplementation was evaluated by Chi-square test. Additional details were added in the Result and the Discussion sections. We did not evaluate sucking problems in this study. We agree that sucking problem might be a potential confounder of breast milk production. This was a limitation of the study and additional discussion was added in the Discussion section. Likewise, we did not evaluate lactation insufficiency in this study. I agree that lactation insufficiency might affect breast milk production. This was a limitation of the study and additional discussion was added in the Discussion section.

Reviewer #3: This is an interesting study that seeks to compare the effectiveness of Wang Nam Yen herbal tea vs. domperidone compared to placebo in achieving improved mother's milk production at 72 hrs following delivery. There were 120 participants in the trial who ranged in delivery gestational age between 28-40 weeks, with a mean GA of ~38 wks and a bw of ~3 kg. The treatment was prescribed for 72 hrs with milk volume measurements via pumped milk expression at each 24 hr interval. There were statistically significant differences between Wang Nam Yen herbal tea and placebo, but not between domperidone and placebo or between Wang Nam Yen herbal tea and domperidone. These are important findings that will serve as the basis for future studies. The following are points that should be addressed to strengthen the clarity and strength of the findings:

Response: Thank you for your compliments. The manuscript was revised as suggested accordingly.

 

Reviewer #3: (1) In the Methods, the placebo tea needs to be delineated and described. How were the participants and investigators blinded given that herbal teas with certain ingredients have particular aromas?

Response: Thank you for pointing it out. Additional detail on placebo tea and tablet was provided in the Methods section. While we intended to produce an ideal placebo tea, the taste and aroma of placebo tea were still different from the Wang Nam Yen herbal tea. However, no participants had a chance to take both active and placebo tea simultaneously. An additional discussion on this limitation was provided in the Discussion section.

Reviewer #3: (2) In the Methods, please describe the block size of randomization.

Response: Thank you for your comment. The block size of randomization and additional detail were added in the Methods section.

Reviewer #3: (3) Methods: in the sample size estimate, you mention milk volume at 4 days but this analysis was at 72 hrs. Perhaps you are using data for power that described milk volume differences at 4 days for your sample size calculation. Please rewrite this section.

Response: Thank you for pointing out a concerning issue. The sample size calculation subheading was revised in the Method section to provide more detail and reproducibility of the study.

Reviewer #3: (4) Methods: it is concerning that women who delivered preterm were analyzed with women who deliver term. I think this is a potential weakness of the study design. Further, in the analyses of neonatal side-effects, the study is not powered to look at preterm neonatal outcomes by treatment. The tea and domperidone may not differ in their safety profile with preterm mothers, but you would need far more mothers enrolled in such a study. I do not think you can extrapolate safety to those born <35wks so easily. You might suggest that this be refined in future studies.

Response: We are sorry for the confusing inclusion criteria and discussion. Initially, we intended to include both preterm and term pregnant women, but they were separated from their child before participation which was one of the exclusion criteria. Hence, all included participants were term pregnant (Gestational age 37-42 weeks). To minimize confusion, the inclusion criteria were amended from the first registered protocol. The inclusion criteria in the Methods section were revised and the reason for inclusion criteria amendment was provided in the Discussion section.

Reviewer #3: (5) The Discussion needs to be checked for accuracy for scientific English to make it more readable. There are grammatical errors in that section that are not seen in prior sections.

Response: Thank you for your comment. The Discussion section was revised was more readable scientific English.

We hope that our responses are satisfactory. Should there be anything that might improve our work, please kindly inform us. Thank you very much for your kind consideration.

Best Regards,

Assoc. Prof. Dr. Krit Pongpirul, MD, MPH, PhD.

On behalf of the authors

---

## [Decision Letter · Decision Letter 1]

30 Jun 2021

PONE-D-21-05057R1

Efficacy of Wang Nam Yen herbal tea on human milk production: A randomized controlled trial

PLOS ONE

Dear Dr. Pongpirul,

Thank you for submitting your manuscript to PLOS ONE. After careful consideration, we feel that it has merit but does not fully meet PLOS ONE’s publication criteria as it currently stands. Therefore, we invite you to submit a revised version of the manuscript that addresses the points raised during the review process.

I have been through the reviewer responses to your feedback and although they are attached I am highlighting below those areas that need to be addressed. Therefore please respond to these 12 points. If you would like to respond to other points made by the reviewers please do. 

Introduction. Please go through the introduction carefully as some parts are still in the past tense while some parts of the methods are in future tense. Sample size calculation. Your response does not really address the concern of the statistical reviewer. It appears that your sample size calculation is based on having 2 parallel groups whereas you have 3. Therefore is the sample size calculation floored? If this is true then it needs to be discussed as a serious limitation in the studyRandomisation Process. The concern is not about who prepared the envelopes, and here you have used an appropriate method, but rather whether the person recruiting patients and deciding whether they met the inclusion criteria know which group the previous patient recruited was allocated to? If they did then they may foreknowledge of what the next allocation will be and this could affect their decision whether to recruit a particular patient. I suggest you describe the recruitment process from this angle? Was recruitment always done by the same clinician? Would they know the previous allocation? Do you need to acknowledge a limitation here?Participants. Inclusion of preterm infants. It is still not clear whether or not you intended to include preterm infants. In the methods it appears that they were not included (you have deleted them) but in the discussion it appears that you intended to include them but indeed there were no preterm infants that met your inclusion criteria. Please clarify. My suggestion is that you leave preterm infants in as an inclusion criteria, but state that none met the inclusion criteria. The discussion could then be about the implications of not having preterms in the studyTable 1. As suggested by the statistical reviewer please remove the statistics. In my view you don’t need all the columns – only the column with totals. However some of the data on the table are not actually baseline characteristics. For example formula use before 24, at 24-48 and 48-72 hours. Consider whether these need to go into a separate table. To avoid confusion please also change the data collection times to hours.Statistical tests. The wording still suggests that both tests (ANOVA and Kruskal Wallace) were used simultaneously. Is it due to the word ‘’and’’. Don’t forget to take the footnote out of Table 1 since you will be taking out the statistical data.Adverse drug effects. It would be helpful to give more information about how adverse drugreactions were sought. Specifically were the participants asked directly about each of the outcomes listed or were they asked a broad question such as do you have any other symptoms. It may be that this is what was listed is just the authors definitions of what would be considered an adverse event. This would clear up the confusion about Table 2. Adverse effects are still not clear from the table. While I understand that you collected adverse events at each 24-hour period the table does suggest there were 3 adverse events in the tea group and 3 in the domperidone group - one presenting in each 24-hour period. This needs to be made clearer.   Methods: I think more detail is needed on the feeding policy of babies after caesarean birth. I note from the methods that the first feeding appears to be at 12-18 hours after birth. Is this correct and if so is this the standard procedure? Most hospitals would start skin to skin and allow breastfeeding either in the operation theatre delivery is under spinal or epidural or as soon as the mother was awake when given general anaesthesia. So for many readers it is unclear whether the baby had been allowed to breastfeed prior to recruitment. Were babies allowed to feed prior to recruitment at 12-18 hours and if so what feeding was allowed? Were mothers allowed to express milk before 12-18 hours?Results: Adding a line into the text to indicate that 'All participants completed their prescribed intervention and were analysed based on their randomization group' will help to reduce future queries on drop-outs. (If this is true)Results: Reviewer 2 has a concern about statistical test results: The authors mention that going by the p values, there is no significant difference between the domperidone and placebo group. However, just looking at the p values does not make logical sense to me. If Wang Nam Yen Herbal tea was better than placebo and Wang Nam Yen Herbal tea is equal to Domperidone, therefore logically Domperidone should be better than placebo. I think it is very misleading to portray that domperidone is similar to placebo as it currently stands. What is the reason for this? Readers could interpret this as selective reporting bias.There are still many grammatical issues right through the manuscript. Reviewer 2 has pointed out a few. Please do get it read by someone with a good grasp of English.

We look forward to receiving your revised manuscript.

Kind regards,

Jacqueline J. Ho, MB.ChB, MMedSc(ClinEpid), FRCP, FRCPCH, FRCPI

Academic Editor

PLOS ONE

Reviewers' comments:

Reviewer's Responses to Questions

**Comments to the Author**

1. If the authors have adequately addressed your comments raised in a previous round of review and you feel that this manuscript is now acceptable for publication, you may indicate that here to bypass the “Comments to the Author” section, enter your conflict of interest statement in the “Confidential to Editor” section, and submit your "Accept" recommendation.

Reviewer #1: (No Response)

Reviewer #2: All comments have been addressed

2. Is the manuscript technically sound, and do the data support the conclusions?

Reviewer #1: No

Reviewer #2: Partly

3. Has the statistical analysis been performed appropriately and rigorously? 

Reviewer #1: No

Reviewer #2: I Don't Know

4. Have the authors made all data underlying the findings in their manuscript fully available?

Reviewer #1: Yes

Reviewer #2: Yes

5. Is the manuscript presented in an intelligible fashion and written in standard English?

Reviewer #1: No

Reviewer #2: Yes

6. Review Comments to the Author

Reviewer #1: Thank you for your previous responses. I think there are some issues not addressed.

As testing at baseline is not correct (there is a substantial literature) please remove as previously requested.

Randomisation: please explain how it was not possible for the envelope to be opened before the front was filled in. Also, with the block size as given, if two allocations are known so is the third and hence there is treatment foreknowledge. Please see the extensive literature on this subject notably by Schultz and Grimes, and the quantification of these biases of Hills, Gray & Wheatley.

Please give more detail no the sample size calculation - the data given is for two groups without Bonferroni correction but there are three here. What was the justification for using the same SD, and presumably effect size.

Are 95% CI for the mean or the variable in the figures?

Reviewer #2: The manuscript is much improved. However, there remains a few areas that still need to be addressed or clarified.

1. Reviewer #2: Outcomes measures: ii. I note that the milk expression is done 2 hours after the baby has last fed. Can the authors clarify how milk collection was done for babies that demand hourly feeding?

Response: The milk collection was done uniformly for all participants. While some babies might demand hourly feeding, all mothers were notified at 2 hours before each milk collection section which allowed the last breastfeeding prior to the milk collection. The Methods section was revised accordingly.

Response to author's response: Thank you for the explanation. However, what the authors have described would mean that the baby must not be breastfed 2 hours before the milk measurement. Therefore, if baby was hungry and crying for feeds during these 2 hours, was he given formula or expressed milk to ease his hunger?

2. Reviewer #2: Outcomes measures: vii. Table 1 has lots of details such as method of payment, number of antenatal visits and presence of labour symptoms. Perhaps the authors could explain why these information were included. If they do not contribute to breast milk production i.e. potential confounders, perhaps they can omit these details so that the table is less cluttered.

Response: Most variables could be potential confounders of breast milk production while some variables were potential confounders for a secondary outcome such as the method of payment, presence of labor symptoms, low socioeconomics status could be potential confounders of postpartum endometriosis. Variable without evidence for potential confounders were omitted from Table 1 as advised.

Response to author's response:

• I still see that method of payment, presence of labor symptoms, low socioeconomics status are still listed in Table 1. The authors explained that these could be confounding factors for the secondary outcomes. However, I still cannot understand how mode of payment and low socioeconomics status can affect any of the secondary outcomes. For example, how can mode of payment cause drug adverse events, jaundice or NICU admission? Could authors also explain how “low socioeconomic group’ causes endometriosis’ and include this somewhere in the text? I cannot understand the scientific rationale behind this.

• On the other hand, ‘highest education level’ was taken out from the table. To me, this is an important confounding factor which should remain in the table-– mothers with higher levels of education could have a better understanding of breastfeeding/ breastfeeding techniques, nutrition etc, thus potentially impact on the amount of breastmilk produced.

3. Reviewer #2: Results: I am confused with the actual number of adverse events. The text mentioned that 3 mothers suffered adverse events and only mentioned 3 types of adverse events, namely postpartum endometriosis, dry mouth and diarrhoea. However, Table 2 reported 6 types of adverse events (headache, dry mouth, diarrhea, muscle cramps, itching, or allergic reactions) and that there was a total of 6 mothers experiencing them. Could the authors clarify?

Response: We apologize for the confusing information. There were 3 mothers who suffered adverse events: one with postpartum endometriosis in the control group (we displayed this event in Maternal complication), one with dry mouth in the domperidone group, one with diarrhea in Wang Nam Yen herbal tea group. Table 2 legend “Drug adverse event includes headache, dry mouth, diarrhea, muscle cramps, itching, or allergic reactions.” was intended to mention which symptoms could be considered as drug adverse event, not the actual drug adverse events occurred.

Thus, Table 2 legend was revised to present actual drug adverse events only.

Response to author's response: Thank you for the explanation

• However, I think the title for Table 2 is still confusing. Suggest to simplify it to just " Table 2: Primary and secondary outcomes"

• I still find there is too much unnecessary details in the table which makes it very difficult to understand and creates a lot of confusion. Would the authors consider only putting the important ones there, and have the rest written in the text?

For example, there was only one mother that had one adverse event – could all of them be grouped under ‘Adverse events’ and then the details that 2 of them occurred at 24 hours which lasted for 72 hours be written in the text instead of having all these information in the table? The way the data is displayed in the table will make the reader think that one participant had an adverse effect occurring at the 24th hour, another at the 48th hour, and another at the 72nd hour, etc.

In addition, presenting the data on Drug adverse events as a row immediately after ‘neonatal complications’ adds to the confusion because these events only happened to mothers and not the babies. Suggest to move the row on adverse events under ‘maternal complications’

• Abbreviation denotated in the legend is difficult to understand. Suggest to follow the style in PLOS. In addition, do you need to include so much detail in the legend? Won’t it be better for the details to be described in the text instead?

4. Reviewer #2: Discussion: ii. Data on confounding factors to milk production should be stated. This includes: when breastfeeding was first initiated (the protocol says that mothers were encouraged to breastfeed within 24 hours, but having her first breastfeed at birth or first breastfeed at 18 hours of life is a big confounder), babies were supplemented with formula in each group, number of infants with sucking problems, number of mothers with lactation insufficiency. If they were not available, they could be discussed as limitations of the study.

Response: We apologize for the confusing information. The first breastfeeding was first initiated after the first dose of intervention which occurred 12-18 hours following delivery. Encouragement to breastfeed their infant within 24 hours of delivery was Sanpasitthiprasong hospital's standard of care policy. The Methods section was revised accordingly. Need for formula milk supplementation was recorded daily and categorized into 4 groups: no formula milk supplement, required 1-day supplement, required 2 days supplement, and required 3 days supplement. Association between intervention groups and formula milk supplementation was evaluated by Chi-square test. Additional details were added in the Result and the Discussion sections. We did not evaluate sucking problems in this study. We agree that sucking problem might be a potential confounder of breast milk production. This was a limitation of the study and additional discussion was added in the Discussion section. Likewise, we did not evaluate lactation insufficiency in this study. I agree that lactation insufficiency might affect breast milk production. This was a limitation of the study and additional discussion was added in the Discussion section.

Response to author's response: Confounders like infant sucking problems, lactation insufficiency have now been adequately addressed. However, I am still confused with the first sentence in your response above " The first breastfeeding was first initiated after the first dose of intervention which occurred 12-18 hours following delivery." Do you mean that the babies were not fed for 12 to 18 hours after birth and had to wait until the mothers were administered the intervention before they could have their first feed? If not, I still think it is important to provide information on when the babies were first put to breast (mean time to first breastfeeding after delivery) or put this as a limitation if the information is not available.

5. Responses to revised manuscript: There are still areas where typo errors/ grammatical errors are found. Eg:

• Method section

Participants were recruited after delivered (‘delivery’ not ‘delivered’) via cesarean section by placing advertisements in four obstetrics wards.

• Discussion section.

Second, while we intended … This might lead to potential bias when some participants might smell or discuss (missing ‘the’) aroma and taste with other participants even no explicit action was observed in this study.

6. Reviewer #2: Results: vii. The authors did mention that milk volume in the domperidone group showed no significant difference from the placebo group after applying Bonferroni correction. Please clarify what was meant by the phrase 'too conservative'. What confounders was used?

Response: An additional detail on Bonferroni correction was added in the Methods and Discussion sections. The disadvantage of Bonferroni's correction was small-than-required adjusted α, thus often caused false-negative result. ‘Too conservative’ is equivalent to ‘weak statistical power’. Potential confounders in this study were maternal age, socioeconomic status, method of payment, gestational age, parity status, adequate antenatal care, presence of labor symptom, previous cesarean section, the experience of breastfeeding, neonatal birth weight, postpartum hemorrhage, fluid balance, and formula milk required.

Response to author's response: The authors mention that going by the p values, there is no significant difference between the domperidone and placebo group. However, just looking at the p values does not make logical sense to me. If Wang Nam Yen Herbal tea was better than placebo and Wang Nam Yen Herbal tea is equal to Domperidone, therefore logically Domperidone should be better than placebo. I think it is very misleading to portray that domperidone is similar to placebo as it currently stands.

7. Reviewer #2: Results: x. The manuscript reported some participants experiencing adverse effects which subsided after discontinuing the intervention. Further details are needed to see how were they related to the intervention or could affect the analysis. How many other participants in each group did not actually complete the intervention as prescribed, ie did not take it 3 times a day for 3 days? At which time point did these participants in each group stopped taking the intervention? Was the milk expression continued at each time point for analysis? If yes, was the participants analysed based on their randomized group? Perhaps these details could be included in the flow diagram or at least in the text of the results.

Response: We apologize for the confusing information. The phrase “which subsided after discontinuing the intervention” means that the adverse symptoms persisted for 72 hours postpartum and then subsided after the trial end (no intervention received). Thus, all participants were complete the intervention as prescribed and analyzed based on their randomization group. The discussion on adverse events was revised in the Discussion section.

Response to author's response: Adding a line into the text to indicate that 'All participants completed their prescribed intervention and were analysed based on their randomization group' will help to reduce future queries on drop-outs.

7. PLOS authors have the option to publish the peer review history of their article (what does this mean?). If published, this will include your full peer review and any attached files.

Reviewer #1: No

Reviewer #2: **Yes: **Wai Cheng Foong

---

## [Author Response · Author response to Decision Letter 1]

19 Aug 2021

Dear Editor,

We thank you and the reviewers for the comments and suggestions. We have tried our best to address and clarify all of the 12 concerning points raised by you and the reviewers. Please find our point-by-point responses below:

Editor: I have been through the reviewer responses to your feedback and although they are attached I am highlighting below those areas that need to be addressed. Therefore please respond to these 12 points. If you would like to respond to other points made by the reviewers please do. 

Response: Thank you for your time and consideration. We sincerely appreciate you and Reviewer 2 for the invaluable advice. The manuscript was revised to address all concerning issues and to meet PLOS ONE’s publication criteria.

Editor: 1. Introduction. Please go through the introduction carefully as some parts are still in the past tense while some parts of the methods are in future tense.

Response: Sorry for improper English grammar use. The introduction and methods sections were thoroughly revised to provide proper English grammar use.

Editor: 2. Sample size calculation. Your response does not really address the concern of the statistical reviewer. It appears that your sample size calculation is based on having 2 parallel groups whereas you have 3. Therefore is the sample size calculation floored? If this is true then it needs to be discussed as a serious limitation in the study

Response: There was a limitation in sample size calculation due to no published peer-reviewed evidence on the effect of this herbal tea. Thus, we calculated the sample size using data based on two parallel groups (domperidone and placebo). This might cause a potential problem due to underpower of the study if the true effect of herbal tea was significantly lower than that of domperidone. On the other hand, if the true effect of herbal tea was significantly higher than that of domperidone, there would be only overestimation of sample size. The Discussion section was revised to address this limitation.

Editor: 3. Randomisation Process. The concern is not about who prepared the envelopes, and here you have used an appropriate method, but rather whether the person recruiting patients and deciding whether they met the inclusion criteria know which group the previous patient recruited was allocated to? If they did then they may foreknowledge of what the next allocation will be and this could affect their decision whether to recruit a particular patient. I suggest you describe the recruitment process from this angle? Was recruitment always done by the same clinician? Would they know the previous allocation? Do you need to acknowledge a limitation here?

Response: Thank you for pointing out this important concern and sorry for the unclear message. The allocation seal envelop opener was the only person who knows which group the patient belongs due to the opener was not involve with the recruitment process, was not the clinician, and was not the data collector as described in the manuscript. This process was done to minimize the potential selection bias from a single block size. The Randomization and blinding section was revised as suggested whereas the limitation of single block size randomization and the method to minimize the potential selection bias was provided in the Discussion section.

Editor: 4. Participants. Inclusion of preterm infants. It is still not clear whether or not you intended to include preterm infants. In the methods it appears that they were not included (you have deleted them) but in the discussion it appears that you intended to include them but indeed there were no preterm infants that met your inclusion criteria. Please clarify. My suggestion is that you leave preterm infants in as an inclusion criteria, but state that none met the inclusion criteria. The discussion could then be about the implications of not having preterms in the study

Response: Sorry for the unclear messages. Yes, we intended to include the preterm infants but ended up with no preterm infants included in the study due to none was met the inclusion criteria as you understood. The Participants and Discussion sections were revised accordingly.

Editor: 5. Table 1. As suggested by the statistical reviewer please remove the statistics. In my view you don’t need all the columns – only the column with totals. However some of the data on the table are not actually baseline characteristics. For example formula use before 24, at 24-48 and 48-72 hours. Consider whether these need to go into a separate table. To avoid confusion please also change the data collection times to hours.

Response: Thank you for your suggestion. The statistics portion of Table 1 was removed as suggested. The data, which were not actually baseline characteristics, were separated to Table 2. The data collection unit was changed from times to hours as advised.

Editor: 6. Statistical tests. The wording still suggests that both tests (ANOVA and Kruskal Wallace) were used simultaneously. Is it due to the word ‘’and’’. Don’t forget to take the footnote out of Table 1 since you will be taking out the statistical data.

Response: Since the Kruskal Wallis test was used for continuous data with non-normal distribution, the Kruskal Wallis test was applied for monthly income and neonatal birthweight in Table 1. After removing the statistics in Table 1 as suggested, the Kruskal Wallis test was not used in remaining statistical tests. Thus, the Statistical analysis section was revised to remove the Kruskal Wallis test accordingly.

Editor: 7 & 8. Adverse drug effects. It would be helpful to give more information about how adverse drug reactions were sought. Specifically were the participants asked directly about each of the outcomes listed or were they asked a broad question such as do you have any other symptoms. It may be that this is what was listed is just the authors definitions of what would be considered an adverse event. This would clear up the confusion about Table 2. Adverse effects are still not clear from the table. While I understand that you collected adverse events at each 24-hour period the table does suggest there were 3 adverse events in the tea group and 3 in the domperidone group - one presenting in each 24-hour period. This needs to be made clearer.

Response: Each of the drug adverse events including headache, dry mouth, diarrhea, muscle cramps, itching, or allergic reactions was asked one by one. The previous data in Table 2 were intended to suggest that one adverse event (diarrhea) in tea group persisted for three days and one adverse event (dry mouth) in domperidone group persisted for three days. We agree that the previous data in Table was not clear enough. Hence, the new Table 3 (previously Table 2) was revised to deliver clear message of adverse event by report the adverse event only at the time of onset to reduce misunderstanding.

Editor: 9. Methods: I think more detail is needed on the feeding policy of babies after caesarean birth. I note from the methods that the first feeding appears to be at 12-18 hours after birth. Is this correct and if so is this the standard procedure? Most hospitals would start skin to skin and allow breastfeeding either in the operation theatre delivery is under spinal or epidural or as soon as the mother was awake when given general anaesthesia. So for many readers it is unclear whether the baby had been allowed to breastfeed prior to recruitment. Were babies allowed to feed prior to recruitment at 12-18 hours and if so what feeding was allowed? Were mothers allowed to express milk before 12-18 hours?

Response: Sorry for miscommunication with the previous information. Both group has similar breastfeeding support for early breastfeeding, frequent breast feeding, correct positioning, and avoiding the use of formula when possible. The standard of care at the Sunpasithiprasong Hospital for early breastfeeding is as early as possible. However, the actual time in some cases might be as late as 24 hours after birth which might not match with the first intervention timepoint in this study. Thus, we promote early breastfeeding as possible and ensure that at least the first breastfeeding was done after the first dose of intervention which occurred 12-18 hours after birth but earlier breastfeeding would be better and there was no restriction on early breastfeeding prior 12-18 hours after birth. The Methods section was revised to clarify the miscommunication.

Editor: 10. Results: Adding a line into the text to indicate that 'All participants completed their prescribed intervention and were analysed based on their randomization group' will help to reduce future queries on drop-outs. (If this is true)

Response: Thank you for your suggestion. This statement was true and was added in the Statistical analysis section accordingly.

Editor: 11. Results: Reviewer 2 has a concern about statistical test results: The authors mention that going by the p values, there is no significant difference between the domperidone and placebo group. However, just looking at the p values does not make logical sense to me. If Wang Nam Yen Herbal tea was better than placebo and Wang Nam Yen Herbal tea is equal to Domperidone, therefore logically Domperidone should be better than placebo. I think it is very misleading to portray that domperidone is similar to placebo as it currently stands. What is the reason for this? Readers could interpret this as selective reporting bias.

Response: Thank you for pointing out this important concern. There was no particular reason for reporting that the domperidone is similar to placebo as the authors also believed that this similarity was due to use of Bonferroni correction, which was conservative method of post-hoc correction, thus we discussed this specific issue on Discussion section. However, the interpretation of result was impartial and followed the proper interpretation of Bonferroni correction. On the other hand, we concerned that if we did not interpret the data as the results presented, that might be another type of selective reporting bias. To be more specific in this study, the effect of herbal vs placebo was slightly higher than effect of domperidone vs placebo, thus it was not uncommon that while herbal vs domperidone was similar, the result of herbal vs placebo and result of domperidone vs placebo might not be the same when the statistical tests applied. This should not be interpreted that the domperidone was similar to placebo since the primary outcome of this study was to study the efficacy of herbal tea vs domperidone and herbal tea vs placebo. The manuscript text about the breast milk per expression at 72 hours postpartum for the domperidone group and the placebo group was deleted to reduce the confusion and to reflect the primary outcome while the result was still provided in the Table 3 to avoid the selective reporting bias.

 

Editor: 12. There are still many grammatical issues right through the manuscript. Reviewer 2 has pointed out a few. Please do get it read by someone with a good grasp of English.

Response: Thank you for your suggestion. The grammatical issues were revised throughout the manuscript accordingly.

We hope that our responses are satisfactory. Should there be anything that might improve our work, please kindly inform us. Thank you very much for your kind consideration.

Best Regards,

Assoc. Prof. Dr. Krit Pongpirul, MD, MPH, PhD.

On behalf of the authors

---

## [Editor Report · Decision Letter 2]

3 Sep 2021

PONE-D-21-05057R2

Efficacy of Wang Nam Yen herbal tea on human milk production: A randomized controlled trial

PLOS ONE

Dear Dr. Pongpirul,

Thank you for submitting your manuscript to PLOS ONE. Unfortunately there are still one or two issues that need to be addressed before the manuscript fully meets PLOS ONE’s publication criteria. Therefore, we invite you to submit a revised version of the manuscript that addresses the points raised during the review process.

You will find your latest version of the manuscript attached with your track changes shown. I have made quite a number of suggestions where the grammar could be improved as well as a couple of suggestions where further information might be needed. 

We look forward to receiving your revised manuscript.

Kind regards,

Jacqueline J. Ho, MB.ChB, MMedSc(ClinEpid), FRCP, FRCPCH, FRCPI

Academic Editor

PLOS ONE
---

## [Author Response · Author response to Decision Letter 2]

10 Sep 2021

Dear Editor,

 Words cannot express how thankful we are with your impressively careful edits of our manuscript. We followed our advise on the language edits and provided responses to your queries.

We hope that the revised version of our manuscript is satisfactory. Should there be anything that might improve our work, please kindly inform us. Thank you very much for your kind consideration.

Best Regards,

Assoc. Prof. Dr. Krit Pongpirul, MD, MPH, PhD.

On behalf of the authors

---

## [Editor Report · Decision Letter 3]

20 Sep 2021

PONE-D-21-05057R3Efficacy of Wang Nam Yen herbal tea on human milk production: A randomized controlled trialPLOS ONE

Dear Dr. Pongpirul,

Thank you for submitting your manuscript to PLOS ONE. The submission looks much better and is very close to meeting the requirements for PLOS ONE.  There is just one small issue and that is that there is no description on how adverse effects were collected. We do not know how the participant was asked. For example was the participant asked whether they experienced any adverse effects and if so what were they or was the participant asked specifically about whether they had experienced any of the listed adverse effects. If so where did this list come from? Were these know adverse effects of the intervention? If they were asked specifically about these was there also a question on whether they experienced any unexpected adverse effects. If they were asked about unexpected adverse effects and reported these what were they?   

We look forward to receiving your revised manuscript.

Kind regards,

Jacqueline J. Ho, MB.ChB, MMedSc(ClinEpid), FRCP, FRCPCH, FRCPI

Academic Editor

PLOS ONE
---

## [Author Response · Author response to Decision Letter 3]

22 Sep 2021

Editor: There is just one small issue and that is that there is no description on how adverse effects were collected. We do not know how the participant was asked. For example was the participant asked whether they experienced any adverse effects and if so what were they or was the participant asked specifically about whether they had experienced any of the listed adverse effects. If so where did this list come from? Were these know adverse effects of the intervention? If they were asked specifically about these was there also a question on whether they experienced any unexpected adverse effects. If they were asked about unexpected adverse effects and reported these what were they?

Response: Many thanks for the thorough assessment and the important comment. Each participant was asked for the presence of each of the potential adverse drug reactions to the study drugs, along with the post-partum clinical parameters, by the study nurse as described in the manuscript. Based on the findings from previous unpublished quasi-experimental study comparing the herbal tea (study group) with a pandan leave tea (control group) (Ref 18), participant in the study group reported no adverse drug reactions whereas one participant in the control group who had consumed the pandan leave tea for two days reported a 5-minute headache episode at 2-3 minutes after consuming the herbal tea on day 3 and day 4. As ‘headache’ could be a general subjective feeling after consuming tea of any kind, we merged the headache in the initial list of potential adverse drug reactions, along with that of domperidone as presented in the Methods section. The participants were also asked about any other symptoms that they experienced after consuming the study drugs that were not in the initial list. In fact, some participants reported a mild degree of abdominal discomfort and sleep difficulty but both symptoms were assessed by the investigators as not related to the study drugs.

---

## [Editor Report · Decision Letter 4]

3 Dec 2021

Efficacy of Wang Nam Yen herbal tea on human milk production: A randomized controlled trial

PONE-D-21-05057R4

Dear Dr. Pongpirul,

We’re pleased to inform you that your manuscript has been judged scientifically suitable for publication and will be formally accepted for publication once it meets all outstanding technical requirements.

Kind regards,

Jacqueline J. Ho, MB.ChB, MMedSc(ClinEpid), FRCP, FRCPCH, FRCPI

Academic Editor

PLOS ONE

Additional Editor Comments (optional):

I have accepted your description about the collection of adverse events. Although the description in the manuscript of how the adverse events were identified could be clearer, (e.g. whether participants were asked specifically whether they experienced each of the listed adverse events or whether they were asked just to describe any adverse events that noticed), I have agreed to accept it as is. 
---

## [Editor Report · Acceptance letter]

19 Jan 2022

PONE-D-21-05057R4 

Efficacy of Wang Nam Yen herbal tea on human milk production:
A randomized controlled trial 

Dear Dr. Pongpirul:

I'm pleased to inform you that your manuscript has been deemed suitable for publication in PLOS ONE. Congratulations! Your manuscript is now with our production department. 

Kind regards, 

on behalf of

Professor Jacqueline J. Ho 

Academic Editor

PLOS ONE